# Multiparametric Ultrasound for Diagnosing Testicular Lesions: Everything You Need to Know in Daily Clinical Practice

**DOI:** 10.3390/cancers15225332

**Published:** 2023-11-08

**Authors:** Carlotta Pozza, Marta Tenuta, Franz Sesti, Michele Bertolotto, Dean Y. Huang, Paul S. Sidhu, Mario Maggi, Andrea M. Isidori, Francesco Lotti

**Affiliations:** 1Department of Experimental Medicine, Sapienza University of Rome, 00161 Rome, Italy; carlotta.pozza@uniroma1.it (C.P.); marta.tenuta@uniroma1.it (M.T.); franz.sesti@uniroma1.it (F.S.); andrea.isidori@uniroma1.it (A.M.I.); 2Department of Radiology, Ospedale Di Cattinara, University of Trieste, Strada di Fiume 447, 34149 Trieste, Italy; bertolot@units.it; 3Department of Imaging Sciences, Faculty of Life Sciences and Medicine, School of Biomedical Engineering and Imaging Sciences, King’s College London, London WC2R 2LS, UK; dean.huang@nhs.net (D.Y.H.); paulsidhu@nhs.net (P.S.S.); 4Endocrinology Unit, Department of Experimental and Clinical Biomedical Sciences “Mario Serio”, University of Florence, 50139 Florence, Italy; mario.maggi@unifi.it; 5Andrology, Female Endocrinology and Gender Incongruence Unit, Department of Experimental and Clinical Biomedical Sciences “Mario Serio”, University of Florence, Viale Pieraccini 6, 50139 Florence, Italy

**Keywords:** ultrasound (US), multi-parametric ultrasound (mp-US), gray-scale ultrasound (GSUS), color-Doppler ultrasound (CDUS), contrast-enhanced ultrasound (CEUS), sonoelastography (SE), testicular lesions, testicular tumors, differential diagnosis

## Abstract

**Simple Summary:**

Testicular lesions (TLs) are challenging clinical or ultrasound findings. When large, hard palpable lumps, TL management is mainly clinical, requiring conventional color-Doppler ultrasound (CDUS) to confirm that they are solid, vascularized lesions suggesting malignancy. However, when their CDUS characteristics are uncertain or when nonpalpable, multiparametric US (mp-US) (i.e., the combination of CDUS and more recent US techniques such as contrast-enhanced US and sonoelastography) plays a key role in their characterization, aimed at differentiating benign from malignant TL. This is relevant, since TLs are frequent, testicular tumors are the most common malignancies in young men, and the accurate assessment of a TL is critical to define its correct management including testicular salvage and US follow-up or orchiectomy. In this scenario, this narrative and pictorial review reports a practical mp-US “identity card” and iconographic characterization of several benign and malignant TLs, useful to the physician in daily clinical practice.

**Abstract:**

Background: Ultrasonography (US) represents the gold standard imaging method for the assessment of testicular lesions (TL). The gray-scale (GSUS) and color-Doppler (CDUS) ultrasound examination allow sonographers to investigate the size, margins, echotexture, and vascular features of TLs with the aim to differentiate benign from malignant lesions. Recently, the use of contrast-enhanced US (CEUS) and sonoelastography (SE) has led to further improvements in the differential diagnosis of TL. Although GSUS and CDUS are often sufficient to suggest the benign or malignant nature of the TL, CEUS can be decisive in the differential diagnosis of unclear findings, while SE can help to strengthen the diagnosis. The contemporary combination of GSUS, CDUS, CEUS, and SE has led to a new diagnostic paradigm named multiparametric US (mp-US), which is able to provide a more detailed characterization of TLs than single techniques alone. This narrative and pictorial review aimed to describe the mp-US appearance of several TLs. Methods: An extensive Medline search was performed to identify studies in the English language focusing on the mp-US evaluation of TLs. Results: A practical mp-US “identity card” and iconographic characterization of several benign and malignant TLs is provided herein. Conclusions: The mp-US characterization of TL reported herein can be useful in daily clinical practice.

## 1. Introduction

Ultrasonography (US) represents the gold standard imaging method for scrotal investigation and is widely used to assess a variety of scrotal diseases [1,2,3]. It is a simple, rapid, and harmless diagnostic tool that is able to provide live images of the scrotal content and, among the imaging techniques, it is the least expensive [1,2,3]. Over time, the use of US has progressively expanded since it is useful to assess scrotal features related to reproductive health, scrotal pain, masses, and trauma [1,2,3].

Currently, conventional gray-scale US (GSUS), supplemented by color-Doppler US (CDUS), is considered as being highly sensitive in detecting testicular lesions, however, it has limits in delineating their nature [3]. If performed by an expert operator, scrotal US, together with clinical history and physical examination, may suggest a differential diagnosis among benign and malignant testicular lesions [4]. However, in some cases, it is difficult to discriminate the benign or malignant origin of a testicular lesion, and in case of a “likely” malignant lesion, it is challenging to suggest a possible cancer type. Hence, to date, histology remains the only certain diagnostic tool to define the nature of a testicular lesion [2].

Recently, the use of contrast-enhanced US (CEUS) and sonoelastography (SE) have led to improvements in the differential diagnosis of testicular lesions [2]. This led to a new diagnostic paradigm, the so called “multiparametric US” (mp-US) [5,6], combining conventional techniques (i.e., GSUS and CDUS) with CEUS [7] and SE [8]. Although not entirely diagnostic, mp-US is able to provide a detailed characterization of testicular lesions [4,9,10]. This is relevant in clinical practice, since an accurate mp-US evaluation of a testicular lesion, beside and along with clinical assessment, is critical to define its correct management including testicular US follow-up or orchiectomy [11]. On the one hand, when “palpable” testicular masses are found, they can be malignant in more than 90% of cases, making radical orchiectomy the standard treatment [12]. On the other hand, when nonpalpable testicular lesions are detected, often incidentally during a scrotal US performed for different reasons (e.g., male infertility, varicocele, history of cryptorchidism, scrotal pain or trauma), the clinical management is more cautious. In fact, these lesions are small and mostly benign [13,14], so unnecessary orchiectomy must be avoided, however, they can also be malignant and can grow over time. In this scenario, US is crucial in the follow-up of small lesions, suggesting surgery in the case of growth/modification of small nodules, especially if testicular tumor-related risk factors (e.g., age 15 to 40 years old, family history of testicular tumors, history of contralateral testicular tumor, cryptorchidism or oligo-/azoospermia) are present [1,15]. Hence, either in the case of palpable testicular masses or, especially, in the case of small testicular lesions, US is useful. In particular, mp-US can help in distinguishing benign and malignant lesions with good accuracy, providing a more detailed characterization than CDUS, CEUS, or SE alone.

The role of mp-US for characterizing testicular lesions has been investigated in some retrospective and prospective studies [16,17,18,19,20], mainly focusing on diagnostic accuracy. This review aims to summarize and update these reports, providing an “identity card” description and a wide iconographic characterization of the GSUS, CDUS, CEUS and SE appearance of several common and uncommon benign and malignant testicular lesions.

## 2. Brief Summary of What to Investigate before Running Mp-US

Clinical history and physical examination are very important to suggest a correct diagnosis when facing a testicular lesion and should be performed before running US.

Anamnesis should investigate testicular malignancy-related risk factors including age (testicular cancer represents the most common malignancy in young men aged 15 to 40 years), family history of testicular tumors, history of contralateral testicular tumor, history of cryptorchidism/orchiopexy, and history of infertility, which represent the main risk factors associated with testicular tumors [1,21]. In addition, previous testicular inflammation (orchitis), torsion, trauma, and other relevant diseases (i.e., Klinefelter syndrome) useful to define a differential diagnosis should be assessed [1,2,3,21]. Patients should be asked to describe eventual signs (testicular mass/nodule, testicular swelling or enlargement, new onset hydrocele, sometimes revealed by self-examination by the patient) and symptoms (i.e., scrotal pain or heaviness, fever, back pain, new onset gynecomastia), together with the moment of onset and their duration [21]. Performing a physical examination before starting US is always recommended: usually palpable hard and large masses are suggestive of testicular tumors while non-palpable lesions are in most cases benign lesions, however, they still need to be assessed carefully [1,2,3,21].

## 3. Mp-US Methodological Standards

Mp-US is increasingly recognized as a valuable problem-solving technique in scrotal pathologies, particularly in differential diagnosis of testicular lesions [9,22]. Mp-US combines conventional techniques (GSUS and CDUS) with CEUS, and SE [9,22], which are relatively recent in evaluating scrotal organs, particularly testicular lesions [2,23].

### 3.1. Scrotal/Testicular Color-Doppler Ultrasonography (CDUS)

The standardization of the methodology used to perform scrotal color-Doppler ultrasonography (CDUS) is relatively new. A detailed description of the standard operating procedures (SOPs) for performing scrotal CDUS have been reported by the European Academy of Andrology for the entire male genital tract [23,24,25,26]. The EAA-proposed SOPs to assess scrotal CDUS and, in particular, testicular lesions, have been reported elsewhere [2,23,25] (see https://www.andrologyacademy.net/eaa-studies (accessed on 20 October 2023)). In particular, testicular US should be performed with a high frequency linear transducer, with the patient in the supine position. A US scan of both testicles should be performed including longitudinal, oblique, and transverse scans, with slow, continuous side-to-side movements that allow for the assessment of the entire parenchyma. The operator should evaluate at GSUS the volume of the testes (using the “ellipsoid” formula [height × width × length × 0.52] for adult testes [27] and the Lambert’s empirical formula [height × width × length × 0.71] for pre-pubertal testes [3]), the echogenicity, the echotexture, the possible presence of testicular calcifications or microlithiasis, and vascularization by CDUS, comparing the two sides. Testicular lesions should be accurately evaluated in longitudinal, oblique, and transverse scans. A complete evaluation should include: (1) diameters (length × height × width); (2) position and extension; (3) type (solid, cystic, mixed), homogeneity (homogeneous/inhomogeneous), and echogenicity (hypoechoic, hyperechoic, anechoic); (4) presence of intralesional calcifications; (5) shape (regular or irregular) and margins (clean-cut, smooth, multi-lobed, infiltrating); (6) vascularization pattern (absent, peripheral, intranodular). The images must be stored to be used for the comparison during follow-up. The report must also describe, besides the lesion, the US characteristics of both testicles and must specify the absence of lesions in the contralateral testicle [1,2,3,23,25].

### 3.2. Contrast-Enhanced US (CEUS)

The methodological standards for the clinical practice of contrast-enhanced US (CEUS) in non-hepatic applications including scrotum investigation have been reported by the EFSUMB Guidelines [27]. As a result, the assessment of some pathological conditions using CEUS has improved [7,27]. Using time–intensity curves, evaluating the wash-in and wash-out curves may help to distinguish malignant from benign tumors, although CEUS analyses still overlap between different histological types [7]. In addition, CEUS can discriminate non-viable regions in testicular trauma and can identify segmental testicular infarction [7,27].

For CEUS, a dedicated machine-setting with a low mechanical index (0.05–0.08) is needed to avoid early microbubble destruction. US contrast medium (very small-sized organic shells filled with gas with high impedance) should be injected as intravenous bolus and followed immediately by 10 mL of 0.9% saline solution. The entire examination needs to be recorded to perform qualitative and quantitative analyses [7].

### 3.3. Sonoelastography (SE)

The methodological standards for the clinical practice of sonoelastography (SE) in non-hepatic applications including testicular investigation have been reported by the EFSUMB Guidelines and Recommendations [28]. So far, strain elastography and shear wave elastography, which includes acoustic radiation force impulse-based techniques, and transient elastography are available. The basic principles of SE have been extensively described in previous EFSUMB Guidelines [29], while methodological standardization for different organs including the testis are reported in the updated EFSUMB guidelines [28]. From a methodological point of view, the use of SE to investigate focal testicular lesions can only be recommended in conjunction with other US techniques as there is overlap between benign and malignant neoplasms [28,30].

## 4. Non-Neoplastic Testicular Lesions

Several non-neoplastic diseases can occur within the testes, and may mimic testicular tumors. Differential diagnosis may be difficult but is imperative to avoid unnecessary surgical interventions. A summary of the clinical characteristics and mp-US features of non-neoplastic testicular lesions is provided in Table 1, and their mp-US appearance is reported in Figure 1, Figure 2, Figure 3, Figure 4, Figure 5, Figure 6, Figure 7, Figure 8 and Figure 9.

### 4.1. Intratesticular Cysts

**Prevalence:** Intratesticular cysts are rare in pediatric patients [31,32] and in young-adult men, while their prevalence in subjects aged >40 years old has been estimated to be 8% to 10% [32].

**Clinical history and physical examination:** Simple intratesticular cysts are usually asymptomatic. They are often incidentally detected during US as they are usually not palpable [33]. However, they can even be palpable, since their size can range from 2 mm to 2 cm [34]. On palpation, they have a soft or tense-elastic consistency.

**GSUS + CDUS:** Upon GSUS examination, intratesticular cysts appear as solitary, or less commonly multiple, anechoic lesions, with a thin, clear, hyperechoic wall, and posterior acoustic enhancement [32]. They often occur near the mediastinum and can be simple or complex (if they have internal septa). Usually, they do not contain solid portions. Only when complicated by an infection or an internal hemorrhage can they appear as hypoechoic or mixed echogenicity lesions [33]. In CDUS, they show absent internal vascularization.

**SE:** Intratesticular cysts generally appear as soft lesions showing a tricolor pattern, blue-green-red [20].

**CEUS:** CDUS is usually sufficient for the diagnosis of a testicular cyst, and CEUS is not necessary. However, if CEUS is performed, intratesticular cysts show absent contrast enhancement [20].

**Differential diagnosis:** Complex testicular cysts must be differentiated from cystic teratomas. Complex teratomas tend to have solid, outlying, vascularized masses, rather than fibrous strands [33]. As a corollary, besides teratomas, cystic areas can be found in embryonal carcinomas, yolk sac tumors, and choriocarcinomas, but they are included in the solid lesion (see below).

### 4.2. Epidermoid Cysts

**Prevalence:** Epidermoid cysts represent 1.5–2.1% of all testicular benign tumors of germ cell origin among men aged 20 to 40 years [35].

**Clinical history and physical examination:** At physical examination, they are palpable painless non-tender nodules (single or multiple) with sizes ranging from 1 to 3 cm [36,37]. Some epidermoid cysts have a tendency to increase in size over time, hence at clinical history and physical examination, they can be described as a firm nodule growing slowly. Very rarely, in post-pubertal subjects, have they been described as associated with/part of part of invasive testicular germ cell tumors, representing a teratoma [37].

**GSUS + CDUS:** In GSUS, testicular epidermoid cysts show a variable appearance depending on their maturation, compactness, and amount of keratin component. They can be classified into four categories: type 1, well-circumscribed rounded lesions with an “onion-ring” pattern consisting of concentric rings of hypoechogenicity and hyperechogenicity (Figure 1, Panel A); type 2, densely echogenic and calcified masses with a dark acoustic shadow; type 3, “target” appearance lesions consisting of a hypoechoic rim with a central area of increased echogenicity; type 4, mixed pattern lesions [38]. The onion-ring pattern, which corresponds to lamellar layers of keratin, is the most typical, accounting for about 60% of cases [39]. CDUS examination shows absent vascularization within the cyst (Figure 1, Panel B).

**SE:** Testicular epidermoid cysts demonstrate hard SE properties, showing low/absent elastic strain [38] (Figure 1, Panel C).

**CEUS:** No contrast enhancement is expected after contrast administration as the lesion is avascular; occasionally it can be present a rim enhancement [38,40] (Figure 1, Panel D).

**Differential diagnosis:** An atypical epidermoid cyst may be mistaken for a malignant tumor, namely embryonal carcinoma with internal necrosis and calcified margins (Figure 2, Panel A), as they can both appear as avascular lesions in CDUS and CEUS (Figure 2, Panel B) and hard in SE. Serum tumor markers can be helpful in differential diagnosis as well as the meticulous study of the margins, which are usually well-demarcated in epidermoid cyst and irregular in malignant tumors. In this scenario, differential diagnosis is decisive, since while the suspicion of a malignant testicular tumor requires orchiectomy, that of an epidermoid cyst, usually benign, avoids the removal of the entire testicle. However, even if epidermoid cysts show a typical “benign” pattern in US, due to the tendency of an increase in size over time both in pediatric and adult patients, and to the rare association with testicular germ cell tumors, they usually are treated with testis-sparing surgery associated with biopsies of the surrounding parenchyma [37].

### 4.3. Testicular Adrenal Rest Tumors (TARTs)

**Prevalence:** TARTs are benign lesions occurring in nearly 40% of patients with congenital adrenal hyperplasia (CAH) [41].

**Clinical history and physical examination:** TARTs are supposed to originate from an adrenal-like pluripotent stem cell type rising from the urogenital ridge, already present in the gonads during embryogenesis, which undergo adrenal differentiation and increased proliferation under the stimulation of high levels of adrenocorticotropic hormone (ACTH) [41]. Generally, TARTs are bilateral and non-palpable due to their occurrence near or within the mediastinum, and a firm mass can be palpated only when the lesion exceeds 2 cm in diameter [41].

**GSUS + CDUS:** In GSUS examination, TARTs usually appear as hypoechoic lesions with irregular, lobulated margins, or less frequently as hypoechoic lesions with hyperechogenic foci, and rarely as hyperechogenic lesions [41] (Figure 3, Panel A). CDUS shows markedly increased intralesional blood flow (Figure 3, Panel B).

**SE:** TARTs usually appear as hard lesions showing low/absent elastic strain [42] (Figure 3, Panel C).

**CEUS:** TARTs show increased contrast-enhancement in CEUS [43,44] (Figure 3, Panel D).

**Differential diagnosis**: It is challenging to discriminate TARTs from other tumors based on their US appearance. However, TARTs are typical findings in patients with CAH. Moreover, their size might decrease with proper glucocorticoid treatment. In addition, TARTs are usually bilateral, an uncommon occurrence in malignant tumors. If small, they can be similar to Leydig cell tumors in GS [45]. Hence, the patient’s clinical history, the occurrence within the mediastinum, and bilaterality of the lesions can help clinicians in the management and appropriate follow-up of these lesions, often avoiding unnecessary orchiectomy [41].

### 4.4. Sarcoidosis

**Prevalence:** Sarcoidosis is a multisystem disease involving the lungs, lymph nodes, kidneys, skin, liver, and spleen, and is characterized by noncaseating granulomas. The reported prevalence of sarcoidosis-related testicular involvement is 4–4.5%, with only 0.5% of symptomatic patients [46].

**Clinical history and physical examination:** Testicular sarcoidosis is usually asymptomatic, being incidentally detected during the patients’ diagnostic work-up [47]. When clinically manifest, testicular sarcoidosis presents as painless or painful nodules [48].

**GSUS + CDUS:** In GSUS, sarcoidosis appears as single or more typically multiple and bilateral small hypoechoic lesions with irregular margins [4,47] (Figure 4, Panel A). In CDUS, testicular sarcoidosis granulomas can show some internal vascular spots [4,49,50] (Figure 4, Panel B).

**SE:** Sarcoidosis granulomas appear as hard lesions showing low/absent elastic strain [4] (Figure 4, Panel C).

**CEUS:** CEUS can confirm the presence of contrast-enhancement within the lesions [4,49,50]; however, a hypovascular appearance of the lesions has been described [51] (Figure 4, Panel D).

**Differential diagnosis:** Differential diagnosis from a testicular neoplasm may be difficult with GS, however, the presence of multiple bilateral lesions involving simultaneously the testis, along with other systemic evidence of sarcoidosis in other organs may suggest the diagnosis [47,48].

### 4.5. Segmental Testicular Infarction

**Prevalence:** Segmental testicular infarction is a rare clinical and US entity [52]. Most cases have been reported as idiopathic; it can also occur as a sequela of recent surgery, inflammatory and infective events, blood disorders such as sickle cell disease and polycythemia, or autoimmune diseases such as vasculitis [52,53].

**Clinical history and physical examination:** Segmental testicular infarction frequently presents with an acute painful, swollen scrotum, especially in men aged 20 to 40 years [54,55]. However, clinically silent cases have been described [52].

**GSUS + CDUS:** In GSUS evaluation, segmental testicular infarction appears as a hypoechoic wedge-shaped lesion [52], usually involving the upper third of the testicle due to poor collateral vessels [56] (Figure 5, Panel A). CDUS shows absent internal vascularization, and a peripheral rim of low vascular signal may be observed [52] (Figure 5, Panel B).

**SE:** It appears in SE as a soft lesion showing high elastic strain [4].

**CEUS:** CEUS can confirm the absence of vascularization within the lesion (Figure 5, Panel C); in cases of subacute testicular infarction, a peripheral hyperenhancing rim can be detected, corresponding to histologic evidence of granulation tissue. During follow-up, the peripheral hyperemic rim diminishes [57].

**Differential diagnosis:** In some cases, the US appearance of segmental testicular infarction can be round-shaped, resembling a testicular tumor [52]. A helpful US feature to distinguish segmental infarction from a testicular tumor is markedly decreased or absent vascular flow in CDUS imaging or in CEUS. In ambiguous cases, the patient’s clinical history and lesion size reduction during follow-up can help the clinician [57].

### 4.6. Abscess

**Prevalence:** Testicular abscess is an unusual finding, complicating 3–5% of epididymitis and epididymo-orchitis [58]. It may also occur as a complication of mumps, trauma, or infarction [36].

**Clinical history and physical examination:** Patients are usually symptomatic, presenting with acute scrotal pain and swelling and frequently with an elevated white blood cell count and fever. Patients often have comorbidities such as diabetes mellitus, human immunodeficiency virus infection, or other immunosuppressive conditions [3].

**GSUS + CDUS:** GSUS appearance is of a focal, complex, heterogeneous low reflecting lesion with irregular margins [48]. In rare cases, focal hyperechoic spots with posterior shadowing may be present, corresponding to gas bubbles within the abscess cavity [59] (Figure 6, Panel A). In CDUS, a hypervascular rim may surround the lesion, with no internal vascular signal [4] (Figure 6, Panel B).

**SE:** Testicular abscess shows in SE a heterogeneous pattern of firmness [4] (Figure 6, Panel C).

**CEUS:** CEUS demonstrates the absence of internal contrast-enhancement with some peripheral enhancement [4,60,61] (Figure 6, Panel D).

**Differential diagnosis:** In some cases, a testicular abscess can resemble a testicular tumor, although it never shows internal vascularization. Evidence of epididymitis/epididymo-orchitis, reactive hydrocele, and scrotal skin thickening could be present in the case of testicular abscess. Serial US examinations to ensure resolution should be performed.

### 4.7. Hematoma

**Prevalence:** Intratesticular hematomas are a possible sequela of a scrotal trauma, which is the third most common cause of acute scrotal pain after epididymo-orchitis and testicular torsion.

**Clinical history and physical examination:** A history of scrotal trauma is usually related to the detection of hematoma in US, even if not all patients report this event [62].

**GSUS + CDUS:** Hematomas in US features change in time according to the evolving of blood products [56]. In the acute phase, hematomas appear as well-circumscribed hyperechoic lesions that subsequently liquefy over time, becoming complex lesions with septa, cystic components, and fluid levels [36,58] (Figure 7, Panel A). Typically, the size of the hematomas decreases over time [3]. In CDUS imaging, there is no signal of internal vascularization [58] (Figure 7, Panel B). It is essential to investigate the vascularization of the residual parenchyma to assess its degree of vitality, and CEUS can be helpful in this context [58,59]. Moreover, CEUS can be useful in ambiguous cases to discriminate between intratesticular hematoma and tumor [63].

**SE:** Intratesticular hematomas show predominantly “soft” SE properties with intermediate/high elastic strain [64] (Figure 7, Panel C).

**CEUS:** CEUS confirms the absence of vascularity within the lesion. Peripheral rim and internal septa enhancement may be present [64] (Figure 7, Panel D).

**Differential diagnosis:** Especially when a scrotal trauma does not occur temporally close to US evaluation, hematomas may mimic testicular tumors [62]. However, performing close, serial US evaluations to assess the decrease in the size of the hematomas can help in the differential diagnosis.

### 4.8. Viral Orchitis and Bacterial Orchitis (Epididymo-Orchitis)

**Prevalence:** The majority of orchitis originate with a previous epididymitis, later on extending to the testis (44–47% of cases). In this case, the etiology is mainly bacterial [1,2]. Conversely, primary orchitis is mainly viral in origin (mumps orchitis), occurring in 20–30% of infected postpubertal men [1,2,35].

**Clinical history and physical examination:** Primary orchitis is less common than epididymo-orchitis and is mostly caused by mumps during or after puberty [35]. Epididymo-orchitis usually follows epididymitis, mainly due to urinary tract infections (e.g., *Escherichia coli*) in young boys and sexually transmitted organisms (e.g., *Naisseria gonorrhoeae* and *Chlamydia trachomatis*) in older patients, although urine cultures are positive in only 10–25% of cases [1,2,3,35]. Clinically, gradual onset of pain (especially in epididymo-orchitis) or acute scrotum can occur. Both primary and secondary orchitis present with painful hemiscrotum and testis enlargement, usually bilateral in the primary form and unilateral in the secondary form, the latter often associated with epididymal enlargement or tenderness or pain [1,2,3]. Scrotal edema, fever and pyuria may occur.

**GSUS + CDUS:** The testis appears enlarged, diffusely hypoechoic and inhomogeneous in GSUS and diffusely hyperemic in CDUS [1,2,3,4,20].

**SE:** In SE, orchitis appears with a heterogeneous pattern of firmness [4,20].

**CEUS:** In CEUS, diffuse vascular hyperenhancement throughout the testis can be observed [4,20].

**Differential diagnosis:** When an enlarged, hard, hypoechoic, and diffusely hyperemic testis is detected in US, differential diagnosis should be considered with large seminomas or lymphomas. The occurrence of bilateral orchitis in postpubertal boys or of concurrent epididymitis in adult men can help to suggest primary or secondary orchitis, respectively, instead of large malignancies.

### 4.9. Idiopathic Granulomatous Orchitis

**Prevalence:** Idiopathic granulomatous orchitis, an inflammatory condition of the testis of unknown etiology, is rarely encountered [65,66]. The condition tends to present in a wide age range (19–84 years), with the highest frequency between 50 and 70 years of age [67].

**Clinical history and physical examination:** Idiopathic granulomatous orchitis is characterized by the presence of non-specific granulomatous inflammation and admixed multinucleated giant cells [65]. Histologically, there is extensive destruction of seminiferous tubules with tubular or interstitial pattern of granulomatous inflammation and prominent collagen fibrosis. Clinical presentation of diffuse granulomatous orchitis includes scrotal pain and testicular enlargement [68].

**GSUS + CDUS:** In GSUS, idiopathic granulomatous orchitis appears as diffusely hypoechoic testis or focal hypoechoic areas with ill-defined margins (Figure 8, Panel A) [69]. CDUS often shows hypervascularization (Figure 8, Panel B) [36].

**SE:** In SE, focal orchitis appears predominantly with a heterogeneous pattern of firmness (Figure 8, Panel C) [8,20].

**CEUS:** In CEUS, diffuse vascular hyperenhancement throughout the lesions can be observed (Figure 8, Panel D) [70].

**Differential diagnosis:** In the case of diffuse orchitis, there is a high suspicion of testicular malignancy, and physical examination fails to differentiate the benign from malignant condition [68]. In this scenario, other signs of inflammation such as scrotal wall thickening and hydrocele may help in the differential diagnosis with testicular tumor [41].

### 4.10. Infectious Granulomatous Orchitis

**Prevalence:** Infective granulomatous orchitis is very rare and can be caused by tuberculosis, brucellosis, and actinomycosis [69]. Tuberculous orchitis usually results from contiguous extension from the epididymis. Infectious granulomatous orchitis can be acute or chronic. In the acute form, patients present with sudden onset of pain, while in the chronic form, they usually present with unilateral scrotal swelling. In some cases, granulomatous orchitis presents as a single or multiple testicular mass and can be suspicious of malignancy.

**Clinical history and physical examination:** Clinical presentation of focal granulomatous orchitis includes acute scrotal pain, fever, and testicular enlargement [68]. Epididymal involvement is common in infectious granulomatous orchitis, especially tuberculosis as well as concurrent septated hydrocele, scrotal wall edema, and calcification of the tunica vaginalis.

**GSUS + CDUS:** In genitourinary tuberculosis, both the GS- and CD-US appearance of the testes can be explained by various pathologic stages of tubercular infection, which include caseous necrosis, granulomas, and healing by fibrosis and calcification [36]. Generally, in GSUS, focal orchitis appears as a single or multiple hypoechoic lesion/s with variable echogenicity and blurred or well-defined margins (Figure 9, Panel A). Vascularization can be internal or peripheral (Figure 9, Panel B).

**SE:** In SE, focal orchitis can appear as both soft and hard lesions [8,20], depending on the stage of infection (Figure 9, Panel C).

**CEUS:** In focal orchitis, CEUS can vary from uniform vascular enhancement throughout the lesions [70] to unenhanced lesions with peripheral rim, depending on the stage of the infection (Figure 9, Panel D).

**Differential diagnosis:** Imaging features of testicular tuberculosis are non-specific and often impossible to distinguish from other more common pathologies such as tumor, infection, inflammation, and infarction [71]. In this scenario, other signs of inflammation such as scrotal wall thickening, hydrocele, and most of all epididymitis, favors the diagnosis of infection [72]. In the suspicion of a tubercular infection, it is mandatory to perform microbiological analysis (e.g., Mantoux test).

## 5. Neoplastic Testicular Lesions

Testicular cancers are rare tumors, accounting for ~1% of adult neoplasms, but represent the most common malignancies in young men aged 15 to 40 years, with increasing incidence rates in many countries in the last two decades [73,74,75].

According to the most recent World Health Organization (WHO) histological classification [76], testicular tumors can be distinguished in two main groups: (1) testicular germ cell tumors (TGCTs), which are the most common (~98% of all testicular cancers), in turn divided into two subclasses, seminomatous (s-TGCTs) and non-seminomatous (ns-TGCTs), and (2) stromal cell tumors, which are rare, even if probably underestimated. In addition, malignancies with testicular localization derived from non-testicular neoplasms (non-primary malignant tumors, i.e., hematologic tumors and metastases) must be considered.

Testicular tumors usually present as painless or paucisymptomatic (heaviness, swelling) testicular masses. In some cases, they are incidentally found during US performed for other reasons.

Serum tumor markers must be included in the diagnostic work-up, namely alpha fetoprotein (α-FP), beta subunit of human chorionic gonadotropin (β-hCG), and lactate dehydrogenase (LDH) [77,78]. Overall, the serum tumor markers show low sensitivity (especially in seminoma) and, if negative, the diagnosis of testicular tumor cannot be excluded [79]. Of note, patients with positive β-hCG often have gynecomastia, since β-hCG is very similar to the LH hormone, which has a direct action in stimulating male breast tissue [80,81].

A summary of the clinical characteristics and mp-US features of neoplastic testicular lesions is provided in Table 2, and their mp-US appearance is reported in Figure 10, Figure 11, Figure 12, Figure 13, Figure 14, Figure 15, Figure 16 and Figure 17.

### 5.1. Seminomatous TGCTs (s-TGCTs)

**Prevalence:** s-TGCTs represent 55–60% of TGCTs. The median age at diagnosis is 20–40 years [82]. They are revealed as components of mixed TGCTs in 30% of cases.

**Clinical history and physical examination:** Patients with s-TGCTs can refer to clinicians for the detection of a testicular firm mass, testicular swelling, testicular pain, or lumbar pain when lymph node metastases are present. However, the diagnosis can be incidental during US performed for other reasons. Infertility [83,84,85] and cryptorchidism [86,87] are common risk factors for seminomas [15]. Physical examination usually reveals a large, hard testicular mass. However, sometimes they are incidental findings in US, since small lesions (<1.5 cm) are not always palpable, especially if placed in the center of the testicle.

**GSUS + CDUS:** The US appearance reflects the histological characteristics that consist of a nest of large, round cells with abundant cytoplasm and distinct borders, with fibrous septa containing lymphocyte infiltration. Occasionally, they can include syncytiotrophoblasts. Necrosis, intercellular edema, and hemorrhage can be present, especially in larger tumors [76,88]. Therefore, in GSUS evaluation, classic seminomas usually appear as focal round homogeneous lesions, hypoechoic to the normal surrounding parenchyma [89,90,91] (Figure 10, Panel A). However, large seminomas can also appear as inhomogeneous lesions with hypo/anechoic internal areas, reflecting tumor necrosis and/or bleeding [92]. Margins can be regular, irregular, or polylobate. Microlithiasis in the affected testicle is common [93,94]. CDUS shows increased peripheral and internal vascularization [89], which is commonly characterized by arborization and branches (Figure 10, Panel B).

**SE:** Seminomas usually appear as hard lesions showing low/absent elastic strain [8,95,96] (Figure 10, Panel C).

**CEUS:** Seminomas usually show hyperenhancement of the whole lesion after CEUS administration, apart from necrotic areas. A rapid wash-in and wash-out are distinctive characteristics of seminomas [40] (Figure 10, Panel D).

**Differential diagnosis:** Several neoplastic and non-neoplastic conditions may mimic testicular seminomas in imaging. Among the non-neoplastic conditions, testicular inflammation including orchitis with or without abscess formation may mimic seminoma. In the acute phase of orchitis, diffuse testicular edema results in the hypoechoic appearance of the testis, which is enlarged compared to the contralateral. Helpful imaging findings to suggest orchitis instead of seminoma include hypoechogenicity (edema) and hypervascularization of the ipsilateral epididymis, reactive hydrocele, associated scrotal edema, and pain [11]. Among the neoplastic conditions, non-seminomatous testicular germ cell tumors (ns-TGCTs) and lymphomas may mimic seminomas. Although seminomas, especially when large, may demonstrate cystic spaces and calcifications, these findings are more commonly encountered in ns-TGCTs. Ns-TGCTs are more likely to have ill-defined margins than seminomas, and age at the diagnosis can help (younger in ns-TGCT, older in seminomas). The US appearance of lymphomas can overlap that of seminomas, but the affected patient population is significantly older [11]. Finally, the imaging appearance of small seminomas can resemble that of Leydig cell tumors (LCTs; see below).

### 5.2. Non-Seminomatous TGCTs (ns-TGCTs)

**Prevalence:** Ns-TGCTs represent 40–45% of TGCTs. They usually occur in younger patients than s-TGCTs (median age at diagnosis 25 years) [82].

**Clinical history and physical examination:** Similar to patients with seminomas, those with ns-TGCTs can refer to clinicians for the detection of a testicular mass, testicular swelling, testicular pain, or lumbar pain when lymph node metastases are present. Due to a faster growth of ns-TGCTs, incidental diagnoses are rare, but possible [97,98].

Gynecomastia is a frequent finding [99]. Serum tumor markers, namely β-hCG and α-FP, are frequently positive, especially when distant metastases are present.

Ns-TGCTs are a heterogeneous group of tumors including different malignancies such as embryonal carcinomas, teratomas, choriocarcinomas, yolk sac tumors, and mixed germ cell tumors, whose mp-US characteristics are reported below.

#### 5.2.1. Embryonal Carcinoma

**Prevalence:** Embryonal carcinoma accounts for about 3% of TGCTs. It represents the most frequent (80%) component in mixed TGCTs.

**Clinical history and physical examination:** See above (Section 5 and Section 5.2).

**GSUS + CDUS:** In GSUS, embryonal carcinoma often appears as an hypoechoic and/or inhomogeneous lesion, frequently with internal calcifications [70] (Figure 11, Panel A). US features may reflect histological features, which consist of significant anaplasia and necrotic areas. Sometimes, these tumors may appear as hypoechoic lesions with calcified margins [100] that may mimic an epidermoid cyst. Focal areas of necrosis and hemorrhage are frequent: in US examination, they appear as anechoic areas (in case of recent hemorrhage) or hyperechoic areas (in case of organized necrosis or hemorrhage). The margins are mainly irregular and polylobulated [101]. CDUS commonly shows increased peripheral and internal chaotic vascularization, even if in a minority of cases they could be avascular if completely necrotic (Figure 11, Panel B).

**SE:** Embryonal carcinomas usually appear as hard lesions showing low/absent elastic strain [102] (Figure 11, Panel C).

**CEUS:** CEUS is usually not recommended for very large lesions with positive serum tumor markers, as embryonal carcinoma usually appears, but can be useful in smaller lesions. Embryonal carcinomas show an inhomogeneous hyperenhancement [103] of the lesion after CEUS administration with rapid wash-in and rapid wash-out (Figure 11, Panel D). However, in rare cases, the lesions may also fail to pick-up the contrast medium [40,103], making it more difficult to diagnose the differential diagnosis (i.e., with atypical epidermoid cyst).

**Differential diagnosis:** The differential diagnosis of embryonal carcinoma is usually with other ns-TGCTs, especially mixed ones due to large size, and is not always possible. Internal calcifications, if present, are usually hallmarks. Embryonal carcinoma with internal necrosis and calcified margins can be mistaken with an atypical epidermoid cyst (see above) [3]. Serum tumor markers can be helpful as well as the meticulous study of the margins, which are usually irregular in embryonal carcinoma and well-demarcated in epidermoid cyst.

#### 5.2.2. Teratoma

**Prevalence:** Teratomas account for about 5–10% of TGCTs.

**Clinical history and physical examination:** See above (Section 5 and Section 5.2).

**GSUS + CDUS:** Teratoma is composed of different somatic tissues, derived from one or more germinal layers (endoderm, mesoderm, and ectoderm). They are usually divided into mature and immature tumors according to histology and in prepubertal and postpubertal according to the age of incidence. Prepubertal tumors are usually benign and have a conservative treatment [104], while postpubertal tumors (both mature and immature) can have malignant attitude and metastasize.

The US appearance varies according to the different histological features of the tumor [90]. They usually appear as well-defined lesions with regular margins. Echotexture can include cystic areas (cystic teratomas) with internal septa, with different content (serous, mucoid, keratinous) [91,105,106]. A differential diagnosis with simple, complex, or epidermoid cyst may sometimes be difficult. Focal calcifications are also common and are mainly due to the presence of cartilage and immature bone tissue [91]. CDUS commonly shows increased peripheral and internal vascularization in the solid portion of the lesion.

**SE:** As other TGCTs, teratomas usually appear as hard lesions showing low/absent elastic strain [102], but depending on the amount of liquid inside, they can also have higher elastic strain [95,102].

**CEUS:** Teratomas show hyperenhancement within the solid part of the lesions with rapid wash-in and rapid wash-out. Anechoic areas are usually non-enhanced.

**Differential diagnosis:** The differential diagnosis of teratoma is usually with other ns-TGCTs, especially mixed ones due to large size and is not always possible. Internal cysts, if present, different content, and internal septa are usually hallmarks.

#### 5.2.3. Choriocarcinoma

**Prevalence:** Choriocarcinoma accounts for about 0.5–1% of TGCTs. They represent about 5–10% of mixed TGCTs.

**Clinical history and physical examination:** See above (Section 5 and Section 5.2). Specifically, choriocarcinomas have a more aggressive attitude compared to other ns-TGCTs, with a higher frequency of blood rather than lymphatic metastases [107]. β-hCG levels are usually very high and therefore they are frequently associated with gynecomastia [91].

**GSUS + CDUS:** Choriocarcinoma can appear as a large, solid inhomogeneous mass, with calcifications and areas with different echogenicity due to necrosis and/or hemorrhages [91,92,101]. However, the GS aspect is not specific and a differentiation from other non-seminomatous tumors is not always easy. In CDUS, peripheral and internal vascularization is highly represented.

**SE:** Choriocarcinomas usually appear as hard lesions showing low/absent elastic strain [102].

**CEUS:** Due to the aggressiveness of the tumor and the frequent positivity of serum testicular markers, the diagnosis can be conducted with GS- and CD-US and it is not necessary to perform CEUS. However, in CEUS, choriocarcinomas show hyperenhancement with rapid wash-in and rapid wash-out.

**Differential diagnosis:** The differential diagnosis of choriocarcinoma is usually with other ns-TGCTs and is not always possible. It could be difficult to distinguish pure forms from mixed ones.

#### 5.2.4. Yolk Sac Tumor

**Prevalence:** Yolk sac tumor is very rare in adults (0–1%) in its pure form while it is the most common TGCT in children (60%). It represents 40% of mixed TGCTs.

**Clinical history and physical examination:** See above (Section 5 and Section 5.2). Of note, serum α-FP is usually high in these tumors [101].

**GSUS + CDUS:** Yolk sac tumors usually appear in CDUS as large, solid inhomogeneous masses, with multiple internal anechoic gaps [91,101]. In CDUS, peripheral and internal vascularization is highly represented.

**SE:** Yolk sac tumors usually appear as hard lesions showing low/absent elastic strain.

**CEUS:** Diagnosis is usually performed with GS and CDUS and it is not necessary to perform CEUS.

However, in CEUS, they show hyperenhancement with rapid wash-in and rapid wash-out.

**Differential diagnosis:** The differential diagnosis of yolk sac tumor is usually with other ns-TGCTs and is not always possible. It could be difficult to distinguish pure forms from mixed ones.

#### 5.2.5. Mixed Germ Cell Tumor

**Prevalence:** Mixed germ cell tumors account for about 20–40% of TGCTs.

**Clinical history and physical examination:** See above (Section 5 and Section 5.2). Of note, mixed germ cell tumors are the most common of the ns-TGCTs and they include the various tumor types described above including the seminomatous and non- seminomatous components, with various percentages within the tumor lesion.

**GSUS**, **CDUS**, **CEUS**, and **SE** reflect the features of the different components and their representation within the lesion (Figure 12, Panel A–D).

### 5.3. Stromal Cell Tumors

**Prevalence:** Stromal cell tumors account for about 3–5% of testicular tumors in adults and 25% in children [104,108]. However, their prevalence is probably underestimated, and according to the recent scientific literature, they represent up to 22% of nonpalpable testicular nodules [109].

**Clinical history and physical examination:** In adults, stromal cell tumors are usually incidental findings detected in US performed for other reasons [110]. Specifically, according to many reports, stromal cell tumors, and in particular Leydig cell tumors (LCTs), are frequent incidental findings in infertile patients [108,111,112]. However, in the case of large tumors, enlargement of the scrotum is reported and can be the first reason for medical consultation.

In children and adolescents, LCTs can lead to precocious puberty due to the excessive androgen production, or gynecomastia, caused by excess estrogen due to androgen aromatization [113]. In adults, excessive androgen secretion is exceptional even in malignant LCTs and is usually not associated with peripheral effects [114]. Conversely, Sertoli cell tumors (SCTs) usually do not show any endocrine activity. In some cases, SCTs are a part of multiple neoplasia syndromes such as Carney complex and Peutz–Jegers [115,116]. Serum tumor markers are always negative in the case of stromal cell tumors and no specifical blood test marker exists for these tumors.

Unlike the TGCTs, the great majority of stromal cell tumors are benign, so testis-sparing surgery is now the standard of care in these tumors [117,118]. In selected patients, a strict radiological surveillance can also be performed [117].

#### 5.3.1. Leydig Cell Tumor (LCT)

**Prevalence:** LCTs account for about 5% of all testicular tumors.

**Clinical history and physical examination:** See above (Section 5.3). Of note, malignancy is reported for 10–15% of LCTs [95]. Histological features of malignancy are cytologic atypia, necrosis, angiolymphatic invasion, increased mitotic activity, atypical mitotic figures, infiltrative margins, extension beyond testicular parenchyma, and DNA aneuploidy [95].

**GSUS + CDUS:** LCTs commonly appear in GSUS as round lesions with homogeneous hypoechoic echotexture and regular well-demarcated margins [40,119]. A hyperechoic halo surrounding the lesion can sometimes be found [120] (Figure 13, Panel A). Dimensions are usually small due to the slow cell growth, and they usually present as single lesions. LCTs are usually unilateral even if, in rare cases, they can involve both testicles [121]. CDUS can show peripheral and, sometimes, intralesional, blood flow [40,119] (Figure 13, Panel B).

**SE:** LCTs usually appear as hard lesions in SE, showing low/absent elastic strain [8,95,96] (Figure 13, Panel C).

**CEUS:** CEUS could be useful for differential diagnosis of LCTs with small seminomas (see below). After CEUS administration, LCT shows a homogeneous and intense hyperenhancement of the whole lesion [122]. A rapid wash-in and a slow wash-out are distinctive characteristics of LCTs [4,40,103] (Figure 13, Panel D). Leydig cells indeed strongly express an angiogenic mitogen, the endocrine gland–derived vascular endothelial growth factor (EG-VEGF) [123]. EG-VEGF may play a role in angiogenesis in LCT growth and therefore in an intense vascularization [122].

**Differential diagnosis:** Distinguishing LCT from small seminomas may sometimes not be straightforward. Nevertheless, differential diagnosis is imperative as the two tumors have a very different clinical course and therefore therapeutic direction [11]. The clinical context is not useful, as age of onset is similar, patients could be asymptomatic in both cases, and infertility could be a risk factor for both tumors. In the case of small lesions, they could be undetectable with clinical examination in both cases but, if palpable, both may have a firm consistency. Serum tumor markers can be negative in both cases. Microlithiasis of the surrounding parenchyma is more frequently identified in seminomas. In GS, both seminomas and LCTs appear as hypoechoic and homogeneous lesions, margins could be well-demarcated in both lesions, and CDUS usually shows internal vascularization in both lesions, even if it can appear more often as intralesional and arborized in seminomas and peripheral in LCTs. SE is similar in LCTs and seminomas as it shows hard lesions with low/absent elastic strain. Hence, CEUS can represent the decisive tool in the differential diagnosis between LCTs and seminomas since the contrast medium diffuses differently in the two lesions. Both seminomas and LCTs are homogeneously hyperenhanced compared to the surrounding parenchyma [20,40,122,124,125,126] with a rapid wash-in [40], while wash-out seems to be different, being slower in LCTs and faster in seminomas [4,40,103]. In addition, according to some reports, LCTs show a greater peak enhancement than seminomas in the wash-in phase [17,103]. These data may depend on the vascular architecture of LCTs, characterized by the high density of regular microvessels [104]. However, the literature does not fully agree on the results of the CEUS kinetics. This depends on very heterogeneous studies, which include different types of lesions and have small sample sizes [127].

Regarding the differential diagnosis between benign and malignant LCTs, no radiological feature can distinguish the nature of the lesion. Hence, although strict radiological surveillance can be performed if a LCT is suspected [98], testis-sparing surgery represents the standard of care in these tumors [117,118], and orchiectomy can be performed in the case of a malignancy in histology.

#### 5.3.2. Sertoli Cell Tumor (SCT)

**Prevalence:** Sertoli cell tumors (SCTs) account for <1% of all testicular tumors, and can be found in men with a wide age range (18 to 80 years), although they are more frequent in young adults [128]. Rarely, SCTs are also reported in pediatric patients [129].

**Clinical history and physical examination:** See above (Section 5.3). Of note, malignancy is reported for 5% of SCTs [97].

**GSUS + CDUS:** SCTs can appear at GSUS as both hypoechoic and hyperechoic lesions, with possible intralesional calcifications (Figure 14, Panel A). Margins are well-demarcated. In some cases, there are large areas of calcification and inhomogeneous echotexture, identifying the so-called “calcifying Sertoli cell tumor”: this specific subtype is usually associated with Carney complex or Peutz–Jegers syndrome [115,116]. CDUS shows a marked internal vascularization of these lesions (Figure 14, Panel B).

**SE:** SCTs usually appear as hard lesions showing low/absent elastic strain (Figure 14, Panel C) [8,95,96].

**CEUS:** SCTs show an homogeneous and intense hyperenhancement of the whole lesion with rapid wash-in and a wash-out similar to the parenchyma [40,103] (Figure 14, Panel D).

**Differential diagnosis:** Differential diagnosis includes LCTs and small seminomas and is sometimes difficult. The kinetic characteristics in CEUS can resemble both LCTs and seminomas, but the relative literature is very scarce as this histotype is rare.

### 5.4. Non-Primary Malignant Tumors

Among the neoplasms not deriving from testicular parenchyma, primary hematologic malignancies of the testes are the most frequent, namely non-Hodgkin lymphoma or primary testicular leukemia. In very rare cases, the testicle can also be the site of metastases [17].

#### 5.4.1. Lymphoma

**Prevalence:** Testicular lymphomas represent 1–9% of all testicular tumors and are frequently B-cell type [130]. They usually affect men older than 50 years.

**Clinical history and physical examination:** Testicular lymphomas can appear as a primary or secondary localization of the disease, and they can be unilateral or bilateral. Patients are usually asymptomatic or paucisymptomatic. Symptoms include, as for other testicular tumors, firm testicular masses, testicular swelling, or testicular heaviness. Specific lymphoma symptoms could be present: fever, weight loss, sweating at night, itching.

**GSUS + CDUS:** In GSUS, lymphomas usually appear as hypoechoic lesions with infiltrating margins (Figure 15, Panel A). The vascular pattern is clearly visible within the lesion and consists of well-organized vessels arranged with a linear, non-branching pattern [17] (Figure 15, Panel B).

**SE:** SE reveals hard lesions showing low/absent elastic strain [17] (Figure 15, Panel C).

**CEUS:** Usually, CEUS shows hyperenhancement of the lesions, with rapid wash-in and wash-out [17], but qualitative and quantitative assessment do not add significant information to conventional CDUS (Figure 15, Panel D).

**Differential diagnosis:** The US differential diagnosis is usually with other testicular tumors that appear hypoechoic at GS (e.g., seminoma). The US hallmark of lymphomas are infiltrating margins and the vascular pattern, which besides the detection in men aged >50 years old can help in suggesting the diagnosis.

#### 5.4.2. Primary Testicular Leukemia

**Prevalence:** Primary testicular leukemia is a rare presentation of leukemia, more frequent in children and young patients. Testicular involvement is found in 1% to 2.4% of boys with acute lymphoblastic leukemia, but is very rare in adult patients [131,132].

**Clinical history and physical examination:** Testicular localization may be simultaneous to the diagnosis of the primary disease or can occur after treatment/remission of the primary disease. Patients are usually asymptomatic. Physical examination can reveal testicular involvement by increased size, irregular swelling, and firm consistency of the testes [133].

**GSUS + CDUS:** GSUS appearance can include two patterns. On the one hand, an infiltrating pattern with irregular hypoechoic longitudinal striae radiating peripherally from the mediastinum to the entire testicle with CDUS showing increased vascularity of non-branching linear patterns has been described. On the other hand, a focal pattern with irregular hypoechoic nodules with smooth irregular margins with increased vascularity in CDUS can be found [17,102,134,135,136] (Figure 16, Panel A). The hypoechogenicity in US reflects the infiltration and aggregation of abnormal lymphoid lesions because the density of tumor cells and vessels is greater than that of normal testicular tissue [17,102,134] (Figure 16, Panel B).

**SE:** In SE, an increased testicular stiffness is reported [102,134] (Figure 16, Panel C).

**CEUS:** Regarding CEUS, lesions appear hyperenhanced due to their high vascularity [17] (Figure 16, Panel D).

**Differential diagnosis:** Primary testicular leukemia can mimic an inflammatory process of the testis such as orchitis. However, the lack of pain and normal appearance of the epididymis can guide the diagnosis [137].

#### 5.4.3. Plasmacytoma

**Prevalence:** Patients with multiple myeloma can rarely present with an intratesticular plasmacytoma. So far, less than a hundred cases have been reported in the literature [138].

**Clinical history and physical examination:** In patients with a known diagnosis of multiple myeloma, intratesticular plasmacytoma should always be suspected. Diagnosis is usually due to a rapid testicular enlargement. The lesion can be hard and elastic at physical examination [139].

**GSUS + CDUS:** Plasmacytoma usually involves the whole testicle, which is enlarged and hypoechoic in GSUS (Figure 17, Panel A), with markedly increased vascularization in CDUS (Figure 17, Panel B).

**SE:** SE reveals a hard lesion showing low/absent elastic strain (Figure 17, Panel C).

**CEUS:** Regarding CEUS, lesions appear hyperenhanced due to their high vascularity [17].

**Differential diagnosis:** Differential diagnosis with orchitis may be difficult, even if the lack of pain and normal appearance of the epididymis can help [140].

#### 5.4.4. Metastases

The testicle is a rare site for the metastatic localization of other tumors. Some literature reports include prostate [141], lung [142], gastrointestinal tumors [143,144], melanoma [145], pancreas [146], kidneys [147], bladder [148], thyroid [149], and neuroblastoma [150].

**Clinical history and physical examination:** In patients with a known diagnosis of extratesticular tumor and detection at palpation of a hard testicular nodule/mass, metastases should always be suspected.

**GSUS + CDUS:** In GSUS, metastases show variable patterns according to the site of the primary tumor. Usually, they present as irregular hypoechoic inhomogeneous lesions vascularized in CDUS [3].

**SE:** SE reveals a hard lesion showing low/absent elastic strain.

**CEUS:** Regarding CEUS, lesions appear hyperenhanced due to their high vascularity.

**Differential diagnosis:** US appearance is not specific; however, metastases are generally found in the setting of widespread disease and are rarely the first reason for presentation [3].

### 5.5. Burned-Out Tumor

**Prevalence:** ‘Burned-out’ testicular tumors are rare clinical entities that describe a spontaneously and completely regressed testicular tumor, which presents at the stage of metastases, in most cases in retroperitoneal lymph nodes, in the absence of clinical or US detection of a testicular nodule [151]. The cause of testicular mass regression is still unknown. Hypotheses of ischemia of the lesion or destruction by the immune system have been advocated [152]. Due to the rarity of burned-out tumors, no specific guidelines exist for diagnosis, clinical, and therapeutic management.

**Clinical history and physical examination:** Symptoms are often non-specific and include nausea, vomiting, and lower back pain due to retroperitoneal lymph-node enlargement, the most common site of metastasis. Retroperitoneal, supraclavicular, cervical, and axillary lymph nodes and, less often, lung and liver localization of metastases can be the first appearance of a ‘burned-out’ testicular tumor. A few patients complain of testicular symptoms [152]. Both seminomas and non-seminomas can have “burned-out” presentations. Commonly, serum tumor markers are very high [152].

**GSUS + CDUS:** No primary testicular lesion is identified, and the tumor is supposed to be reduced to a fibrotic scar, represented by a linear macrocalcification (>0.2–0.3 cm) with a rear shadow cone [153,154]. Histologically, it corresponds to psammoma bodies (smooth laminated intratubular calcifications) and hematoxyphilic bodies (non-laminated intratubular calcifications) [155,156,157,158]. Occasionally, signs of burned-out tumors are represented by hypoechoic irregular areas within the testicle with scarce vascularization [155,156,157,158]. Testicular atrophy and microlithiasis have also been reported in relation to burned-out tumors [155,156,157].

**SE:** In the parenchyma surrounding the scar/calcification, a focal area of increased stiffness can be observed in SE.

**CEUS:** The fibrotic scar and surrounding areas are usually not enhanced with CEUS [125].

**Differential diagnosis:** A burned-out tumor can be confused with a simple fibrotic scar or a linear macro-calcification. When the imaging is uncertain, serum tumor markers, which show high levels in burned-out tumors, must be performed to exclude distant spread of the disease.

## 6. Mp-US: Advantages, Limitations, and Future Perspectives

According to the aforementioned information, mp-US shows relevant advantages in relation to the investigation of testicular lesions. In fact, combining different US techniques, mp-US can provide a more detailed characterization of testicular lesions than a single US technique alone, helping in the differential diagnosis of benign or malignant lesions and in the effort to identify the type of lesion assessed [1,2,3]. However, mp-US also has some limitations. In some cases, even using mp-US, it can be difficult to discriminate the benign or malignant origin of a testicular lesion, and in the case of a “likely” malignant lesion, it is challenging to suggest a possible cancer type. Hence, to date, histology remains the only certain diagnostic tool to define the nature of a testicular lesion. In addition, while GS and CDUS are often sufficient to suggest the benign or malignant nature of a testicular lesion [1,2,3], and CEUS can help in better defining the nature of a lesion [7,29], SE added value in clinical practice remains to be proven [8,30,32] and, so far, its use in increasing the diagnostic accuracy of other US techniques is poor. Regardless, future perspectives of the imaging of testicular lesions are promising. Mp-US, eventually associated with other imaging techniques (e.g., magnetic resonance imaging), and the technical advancement of US devices will help to characterize testicular lesions more and more, with the aim to identify the nature of a testicular lesion with increasing accuracy. In addition, mp-US, eventually associated with other imaging techniques, will try to identify parameters suggesting an early tumor in men with testicular malignancy risk factors (e.g., cryptorchidism). In our opinion, in a few years, the new diagnostic paradigm will be “multiparametric imaging”, combining more and more sophisticated US techniques and devices with other imaging techniques, in the attempt to increase the diagnostic accuracy of testicular lesions as much as possible.

## 7. Conclusions

Mp-US is a valuable diagnostic paradigm combining information derived from different US techniques (GSUS, CDUS, CEUS, and SE), which, along with clinical history and physical examination, can help in the differential diagnosis of testicular lesions. Mp-US can provide a more detailed characterization of testicular lesions than a single US technique alone. Although GS and CDUS are often sufficient to suggest the benign or malignant nature of testicular lesions, CEUS can be decisive in the differential diagnosis of unclear findings, and SE can help to strengthen the diagnosis. The knowledge of mp-US patterns of testicular lesions, summarized in this review, is useful to the physician in daily clinical practice to discriminate benign and malignant lesions, thus improving the management of critical patients by suggesting testicular salvage and US follow-up or orchiectomy.

## Figures and Tables

**Figure 1 cancers-15-05332-f001:**
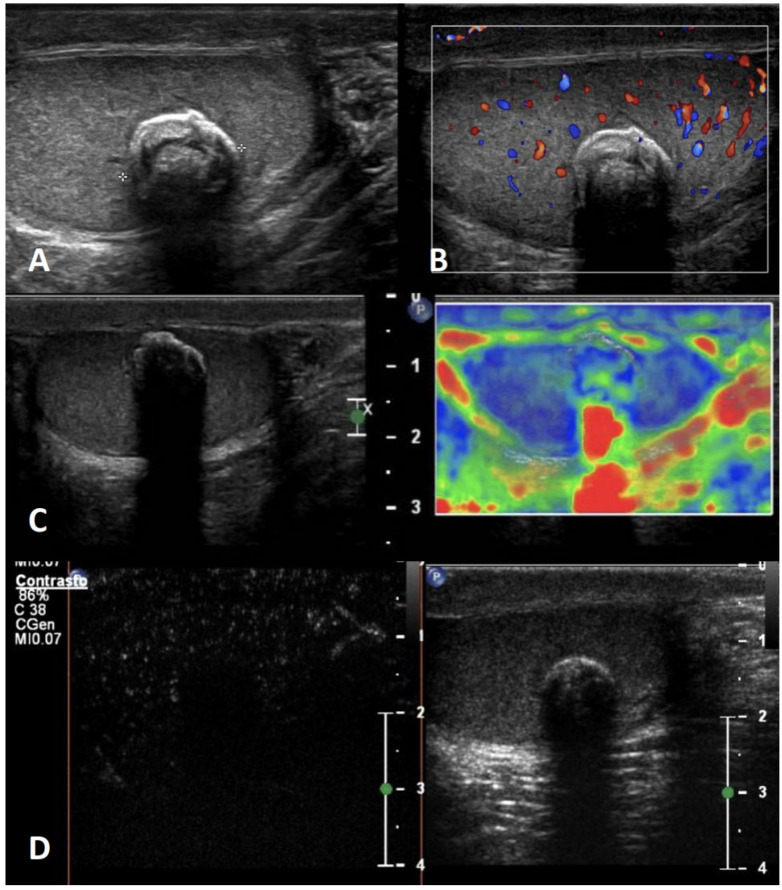
Epidermoid cyst. GSUS demonstrates a well-circumscribed, solid, mixed-reflectivity lesion with high-reflectivity “onion-skin” peripheral rims (panel **A**), avascular in CDUS (panel **B**) in a 17-year-old male patient who was referred for testicular pain. SE shows a mixed elasticity lesion (panel **C**), demonstrated by a blue-green pattern, while contrast-enhanced US demonstrates a clear lack of enhancement within the lesion (panel **D**).

**Figure 2 cancers-15-05332-f002:**
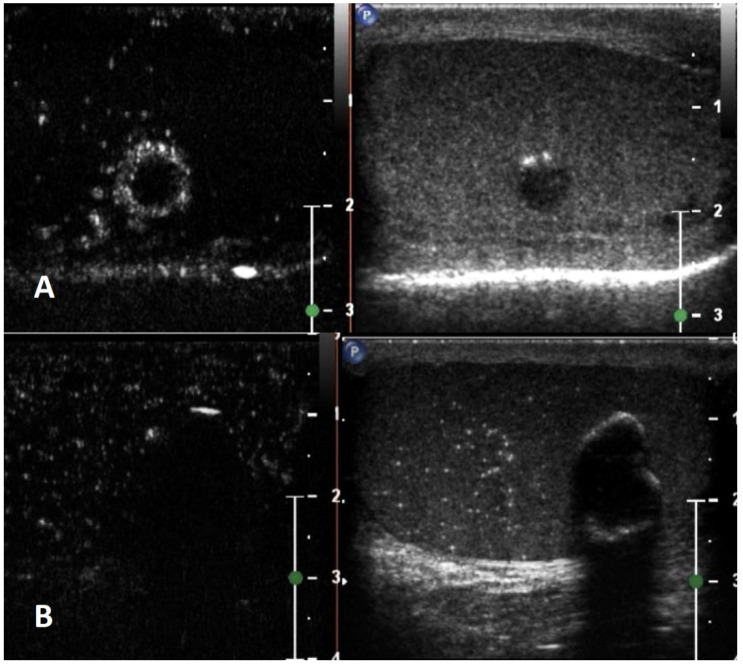
Embryonal carcinoma with internal necrosis (panel **A**) in a 30-year-old man referred for varicocele and atypical epidermoid cyst (panel **B**) in a 16-year-old boy referred for a lump in the testis: both demonstrate in CEUS a lack of vascularity.

**Figure 3 cancers-15-05332-f003:**
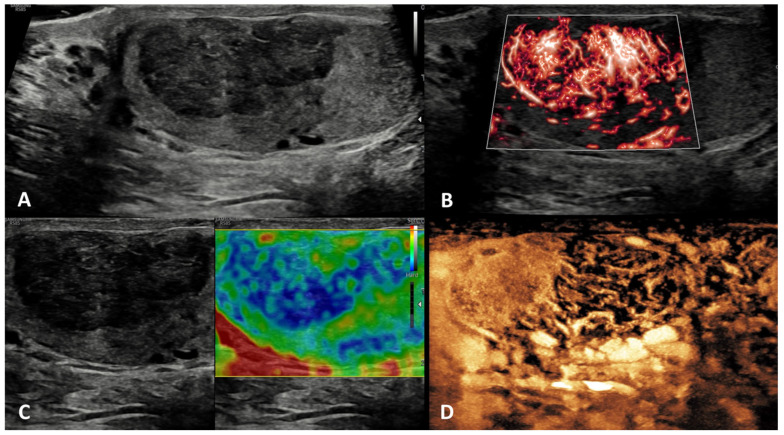
Testicular adrenal rest tumor. GSUS (panel **A**) and CDUS (panel **B**) demonstrated bilateral hypoechoic lesions, highly vascularized, with irregular, lobulated margins in a 28-year-old man with congenital adrenal hyperplasia. In SE, they appeared as hard lesions (panel **C**). TARTs showed increased contrast-enhancement in CEUS (panel **D**).

**Figure 4 cancers-15-05332-f004:**
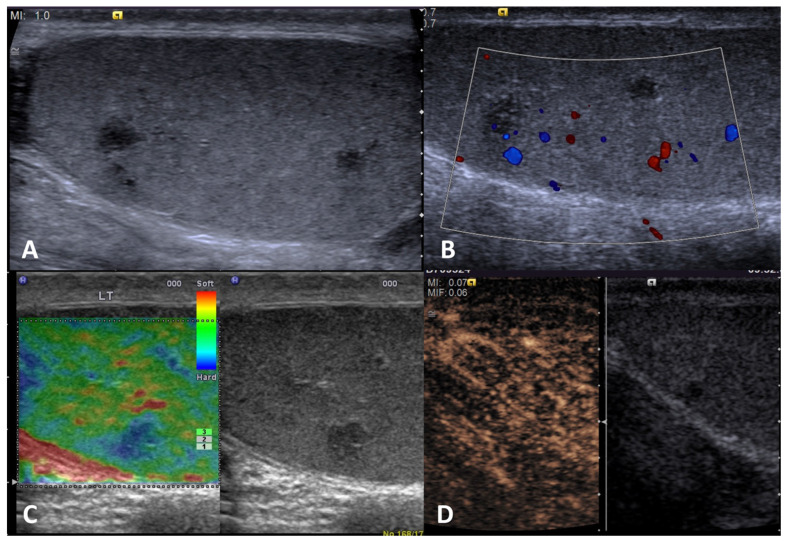
Sarcoidosis. GSUS (panel **A**) and CDUS (panel **B**) demonstrated multiple small hypoechoic lesions with irregular margins and some internal vascular spots. In SE, sarcoidosis granulomas appeared as hard lesions (panel **C**). CEUS can confirm the presence of contrast-enhancement within the lesions (panel **D**).

**Figure 5 cancers-15-05332-f005:**
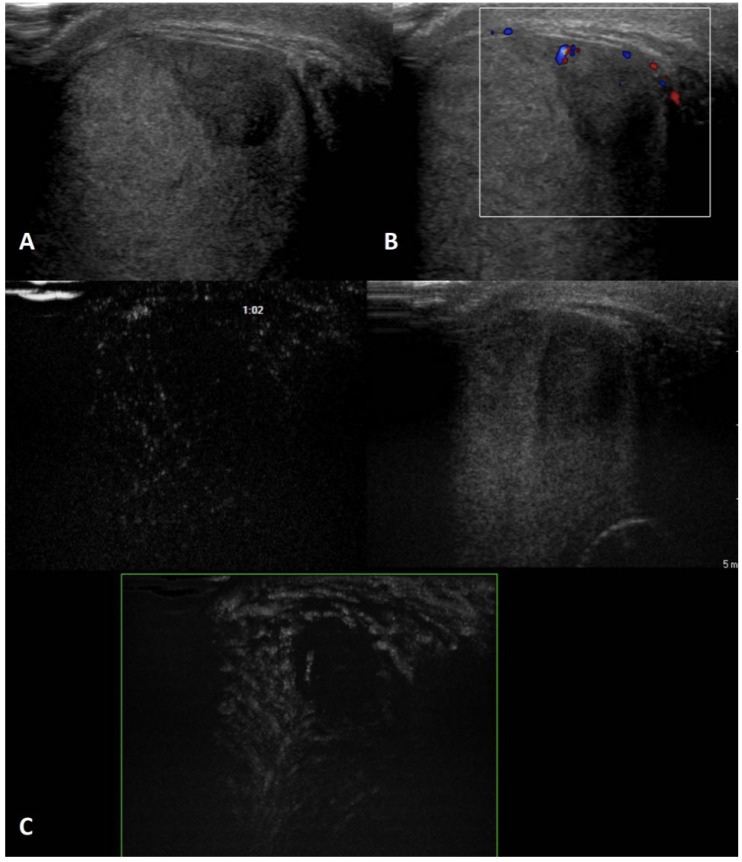
Segmental testicular infarction. GSUS demonstrates hypoechoic lesions, mimicking a tumor (panel **A**) in a 28-year-old patient with a positive personal history of testicular cancer who was performing regular US follow-up. CDUS shows a lack of internal vascularization (panel **B**). CEUS confirmed the absence of vascularity within the lesion (panel **C**).

**Figure 6 cancers-15-05332-f006:**
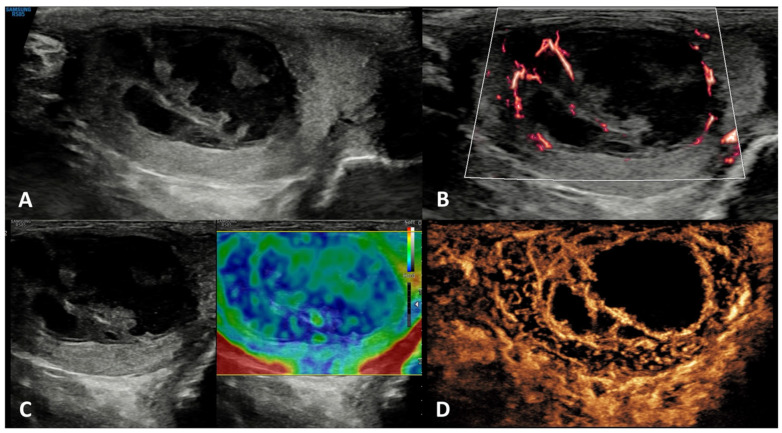
Abscess. GSUS demonstrated a focal, complex, heterogeneous low reflecting lesion with irregular margins (panel **A**). CDUS showed a hypervascular rim surrounding the lesion, with no internal vascular signal (panel **B**). In SE, testicular abscess showed a heterogeneous pattern of firmness (panel **C**). CEUS demonstrated the absence of internal contrast-enhancement with some peripheral enhancement (panel **D**).

**Figure 7 cancers-15-05332-f007:**
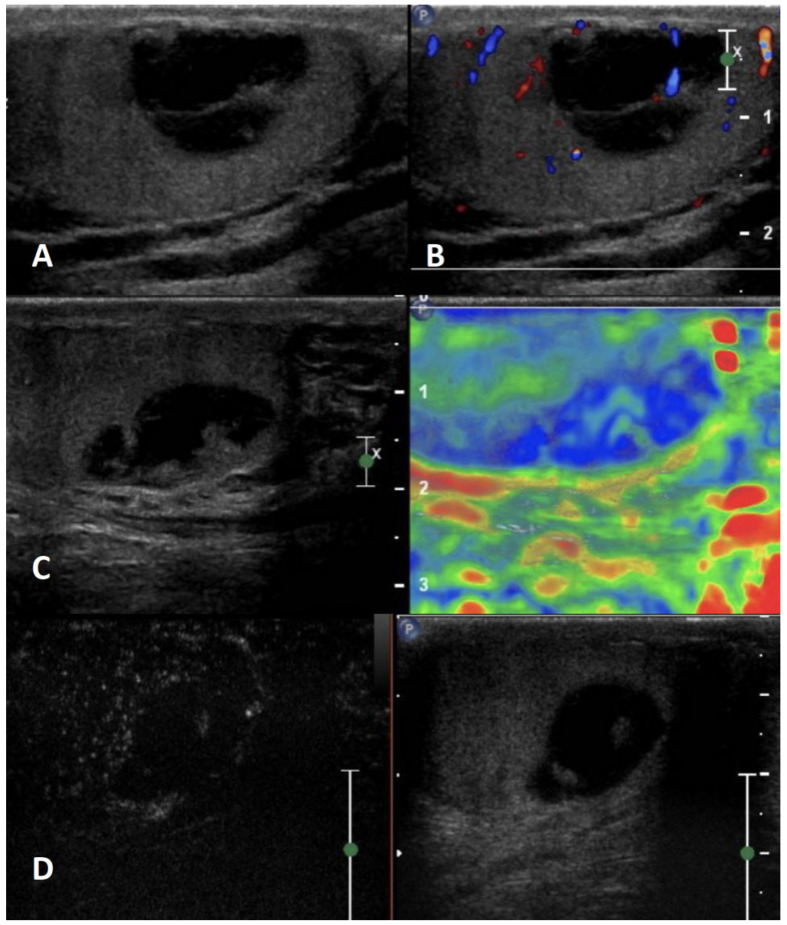
Hematoma. GSUS demonstrated well-circumscribed anechoic lesions with septa and solid components (panel **A**) in a 38-year-old man referred after testicular trauma related to the ball of padel. CDUS showed a lack of internal vascularization (panel **B**). In SE, hematoma showed intermediate/high elastic strain (panel **C**), whereas CEUS confirmed the absence of vascularity within the lesion (panel **D**).

**Figure 8 cancers-15-05332-f008:**
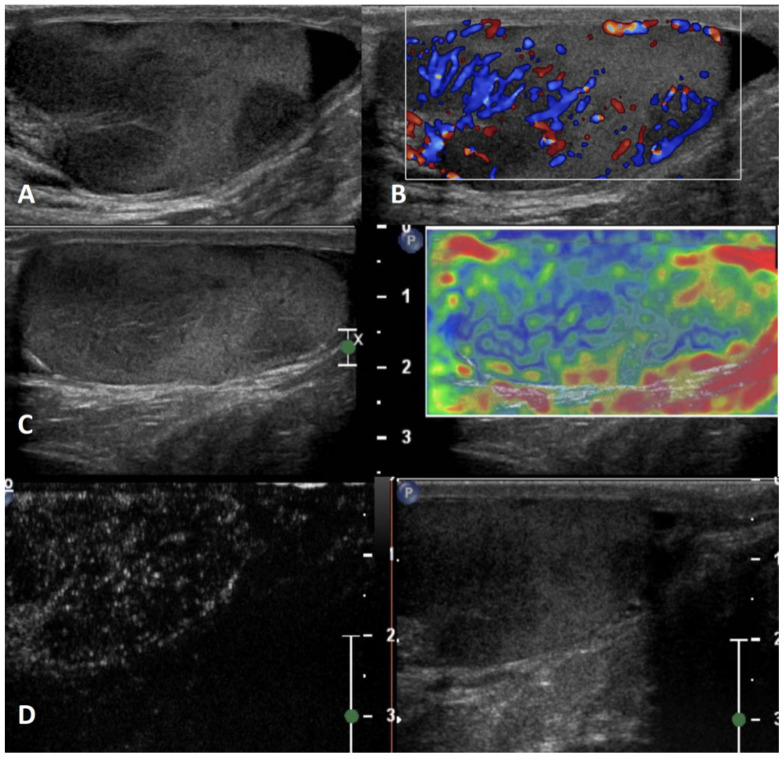
Idiopathic granulomatous orchitis. GSUS demonstrated multiple ill-defined, homogeneous, hypoechoic lesions (panel **A**) in a 24-year-old patient with a positive personal history of testicular cancer (seminoma) diagnosed 6-months earlier during his regular US follow-up. CDUS showed increased internal vascularization (panel **B**). In SE, the testis showed diffuse intermediate elastic strain (panel **C**), whereas CEUS confirmed the hyperenhancement within the lesions (panel **D**).

**Figure 9 cancers-15-05332-f009:**
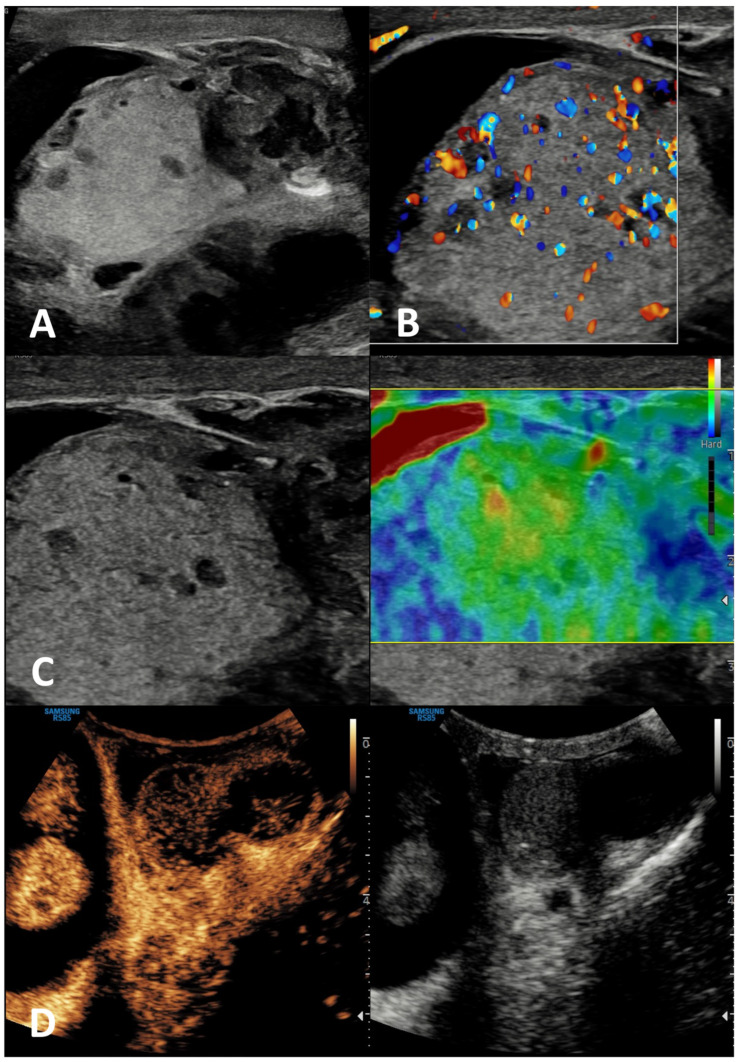
Tuberculous granulomatous orchitis. GSUS demonstrated focal hypoechoic lesions with blurred margins (panel **A**). CDUS showed only peripheric vascularization (panel **B**). In SE, tuberculous granuloma showed intermediate elastic strain (panel **C**). CEUS confirmed the hypoenhanced lesions with peripheral rim (panel **D**).

**Figure 10 cancers-15-05332-f010:**
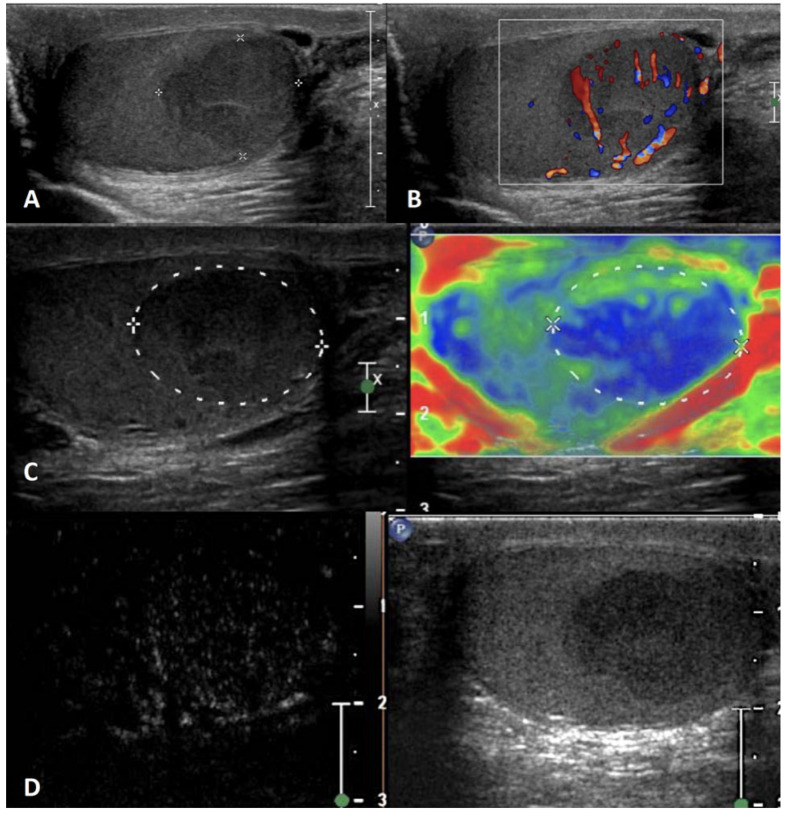
Seminoma. GSUS demonstrated a well-circumscribed homogeneously hypoechoic lesion in a 37-year-old man referred for infertility (panel **A**). CDUS showed increased internal vascularization (panel **B**). In SE, seminoma showed absent elastic strain (panel **C**), whereas CEUS confirmed the hyperenhancement within the lesion (panel **D**).

**Figure 11 cancers-15-05332-f011:**
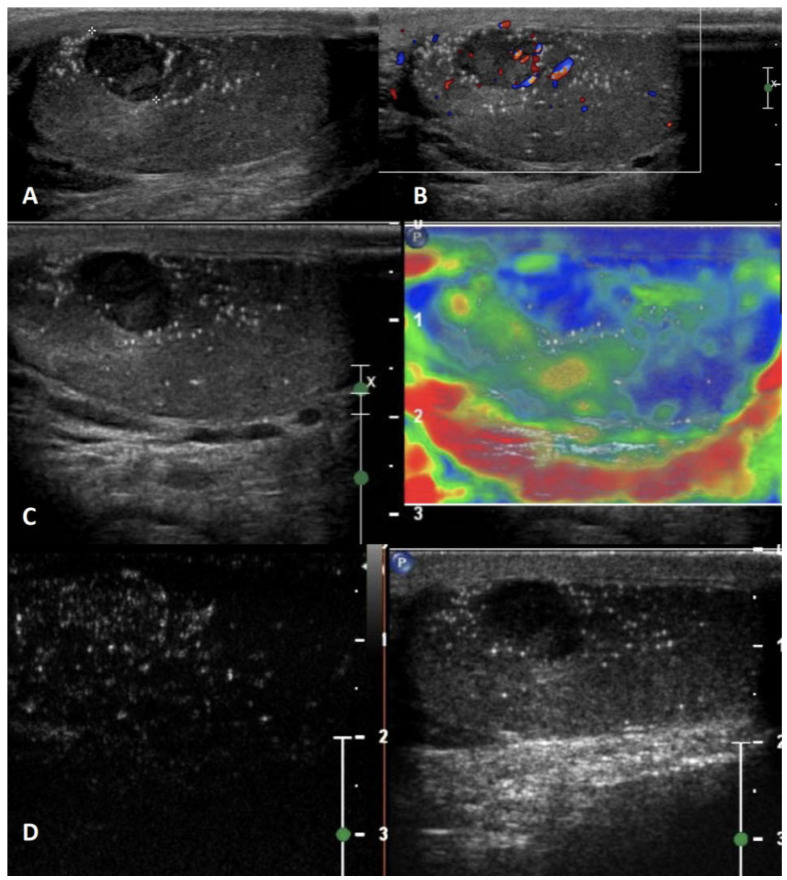
Embryonal carcinoma. GSUS demonstrated a markedly hypoechoic lesion (panel **A**) in a testis with starry sky appearance in a 29-year-old patient referred for testicular pain in the contralateral testis. CDUS showed peripheral and internal vascularization (panel **B**). In SE, the tumor showed absent elastic strain (panel **C**), whereas CEUS confirmed the hyperenhancement within the lesion (panel **D**).

**Figure 12 cancers-15-05332-f012:**
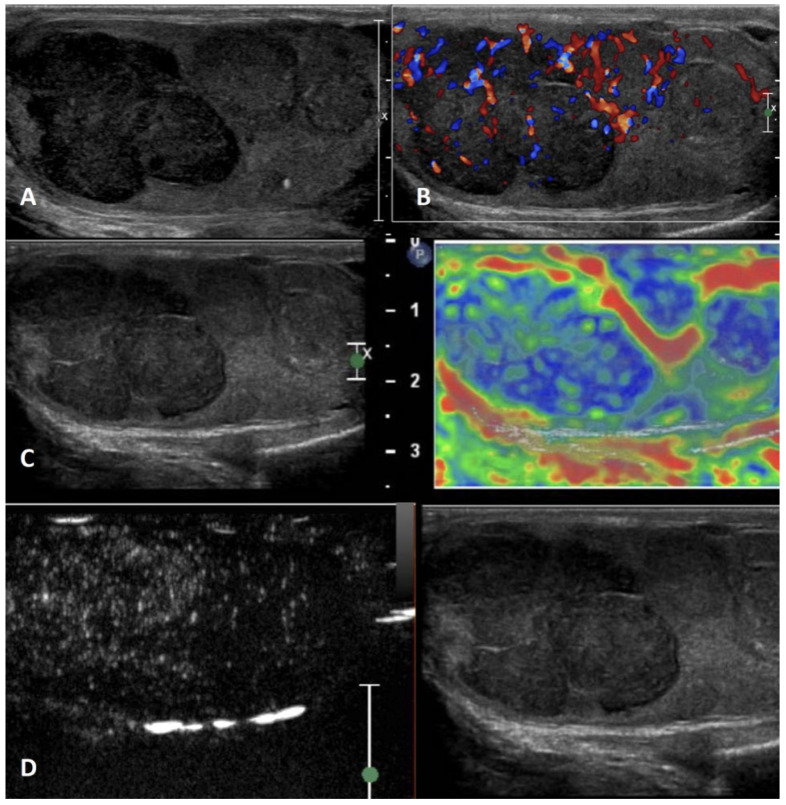
Mixed germ cell tumor. GSUS demonstrated multiple markedly and mild hypoechoic lesions (panel **A**), occupying almost the entire testis of a 26-year-old patient referred for scrotal swelling. CDUS showed peripheral and markedly internal vascularization (panel **B**). In SE, the tumor showed intermediate/absent elastic strain (panel **C**). CEUS demonstrated hyperenhancement of the entire lesion (panel **D**).

**Figure 13 cancers-15-05332-f013:**
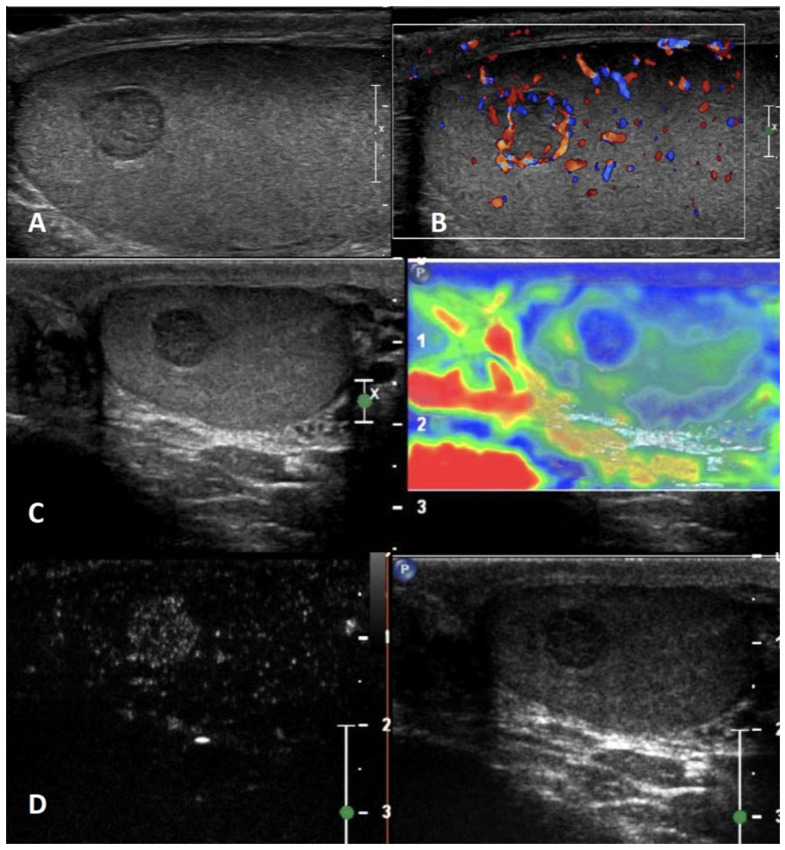
Leydig cell tumor. GSUS demonstrated a well-defined hypoechoic lesion (panel **A**), with a hyperechoic halo in a 34-year-old man referred for primary infertility. CDUS showed peripheral and marked internal vascularization (panel **B**). In SE, the tumor showed an absent elastic strain (panel **C**). CEUS confirmed the hyperenhancement within the lesion (panel **D**).

**Figure 14 cancers-15-05332-f014:**
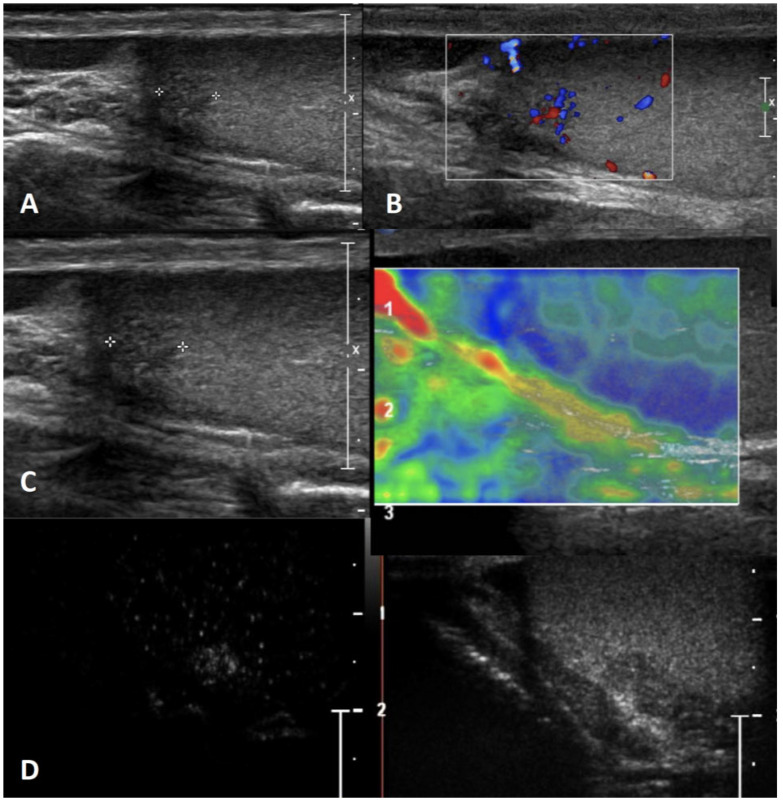
Sertoli cell tumor. GSUS demonstrated a mild hypoechoic lesion (panel **A**) with irregular margins in a 46-year-old patient referred for hypogonadism. CDUS showed markedly internal vascularization (panel **B**). In SE, the tumor showed absent elastic strain (panel **C**). CEUS confirmed the enhancement within the lesion (panel **D**).

**Figure 15 cancers-15-05332-f015:**
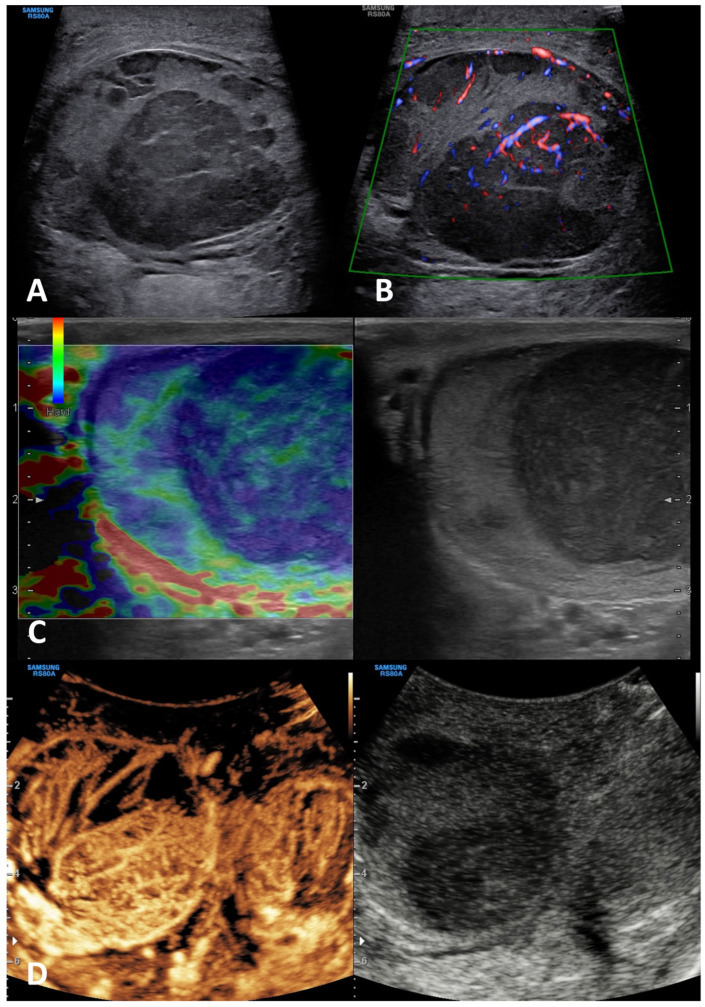
Lymphoma, nodular pattern. GSUS demonstrated a markedly hypoechoic lesion, with a multinodular aspect (panel **A**), with irregular margins, and interesting epididymis tail. CDUS showed markedly internal vascularization (panel **B**). In SE, the tumor showed an absent elastic strain (panel **C**). CEUS showed hyperenhancement of the lesions, with rapid wash-in and wash-out (panel **D**).

**Figure 16 cancers-15-05332-f016:**
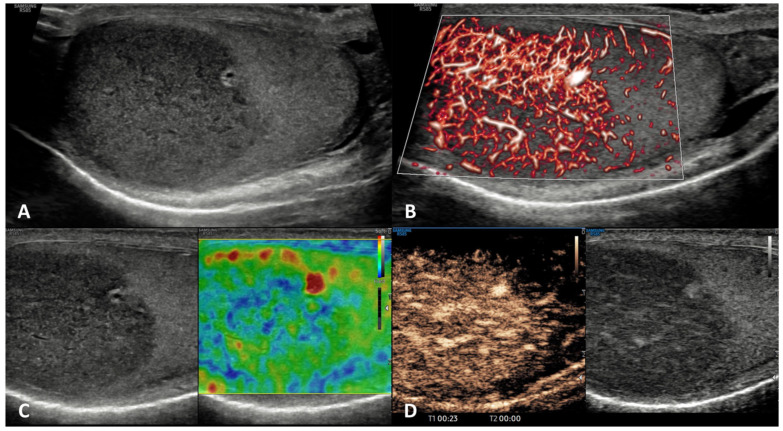
Leukemia. GSUS demonstrated a hypoechoic lesion (panel **A**) with regular margins. CDUS showed internal vascularization of the lesion (panel **B**). In SE, the lesion demonstrated an intermediate/soft elastic strain (panel **C**). In CEUS, the lesion appeared hyperenhanced due to its high vascularity (panel **D**).

**Figure 17 cancers-15-05332-f017:**
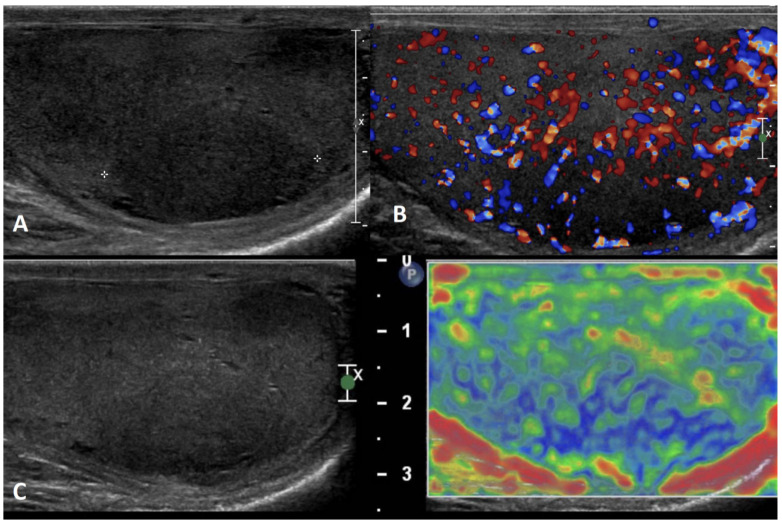
Plasmacytoma. GSUS demonstrated multiple, both mild and markedly hypoechoic lesions (panel **A**) with smooth margins in a 72 year-old-man referred for scrotal swelling, with a positive personal history of plasmacytoma. CDUS showed internal vascularization of the lesions and hypervascularization of the entire testis (panel **B**). In SE, the lesions demonstrated intermediate elastic strain (panel **C**).

**Table 1 cancers-15-05332-t001:** Ultrasound, CDUS, and CEUS characteristics of principal non-neoplastic intratesticular lesions.

Non-Neoplastic Intratesticular Lesions					
	Clinical Presentation	GS-US	CD-US	CEUS	SE
Simple cyst	Asymptomatic/incidental finding, usually not palpable	Rounded anechoic lesions with thin, clear, hyperechoic wall and posterior acoustic enhancement	Avascular	Unenhanced	Soft lesion with High elastic strain
Epidermoid cyst	Asymptomatic can be palpable	Well-circumscribed rounded lesion with “onion ring” aspect (concentric hypo- and hyper-echoic rings) OR densely calcified mass with acoustic shadow OR cyst with hypoechoic rim and central calcification OR mixed atypical pattern	Avascular	Unenhanced/ Perilesional Rim enhancement	Hard lesion with low/absent elastic strain
Adrenal rest tumor	Patients with congenital adrenal hyperplasia; usually not palpable	Hypoechoic lesions with irregular margins, hyperechogenic foci, typically localized in the mediastinum testis, usually bilateral	Markedly vascularized	Hyperenhanced	Hard lesions with low/absent elastic strain
Sarcoidosis	In the context of a multisystem disease; granulomas in other organs; asymptomatic OR painless/painful mass	Hypoechoic lesions with irregular margins, often bilateral	Possible signs of internal vascularization	Hypoenhanced	Hard lesions with low/absent elastic strain
Segmental infarction	Idiopathic or consequent to surgery, inflammatory events, blood disorders or autoimmune diseases; usually acute painful swollen scrotum OR asymptomatic	Hypoechoic wedge-shaped or roundish area	Avascular OR peripheral rim of low CD	Unenhanced/ perilesional rim enhancement	Soft lesions with high elastic strain
Abscess	Acute scrotal pain and swelling/ fever/high WBC	Complex heterogeneous low reflecting lesion with irregular walls (in rare cases focal hyperechoic spots due to gas bubble)	Avascular/ vascular rim	Unenhanced/ perilesional rim enhancement	Heterogeneous pattern of firmness
Hematoma	History of scrotal trauma	Well-circumscribed hyperechoic lesions which subsequently liquefy over time, becoming complex lesions with septa, cystic components, and fluid levels. Size decrease over time.	Avascular	Unenhanced/ perilesional rim enhancement	Soft lesion with intermediate/high elastic strain
Idiopathic (diffuse) granulomatous orchitis	In the context of a multisystem disease; asymptomatic OR painless/painful mass	Diffusely hypoechoic testis or hypoechoic areas with ill-defined margins	Markedly vascularized	Hyperenhanced	Heterogeneous pattern of firmness
Infectious (focal) granulomatous orchitis	Acute scrotal pain, testicular enlargement, fever; possible epididymal enlargement, scrotal wall thickening and hydrocele	Single or multiple variable echogenicity areas with blurred margins; appearance depends by the pathologic stages of infection, which include caseous necrosis, granulomas, and healing by fibrosis and calcification	Internal OR peripheral depending on the stage	Unenhanced/ perilesional rim enhancement OR hyperenhanced	Heterogeneous pattern of firmness depending on the stage

**Table 2 cancers-15-05332-t002:** Ultrasound, CDUS, and CEUS characteristics of principal neoplastic intratesticular lesions.

Neoplastic Intratesticular Lesions						
	Clinical Presentation	Serum Tumor Markers	GS-US	CD-US	CEUS	SE
Leydig cell tumor	Generally asymptomatic; it can produce androgens	Negative	Hypoechoic, homogeneous well-demarcated lesion (possible hyperechoic halo)	Hypervascularized	Homogeneously hyperenhanced (rapid wash-in, delayed wash-out)	Hard lesions with low/absent elastic strain
Sertoli cell tumor	Asymptomatic; they can be a part of multiple neoplasia syndromes such as Carney complex and Peutz–Jegers	Negative	Hypo- or hyper-echoic lesion, with possible calcifications	Hypervascularized	Homogeneously hyperenhanced	Hard lesions with low/absent elastic strain
Seminoma	Testicular swelling, pain, lumbar pain OR asymptomatic palpable firm testicular mass; possible gynecomastia	possible increase of β-hCG	Hypoechoic homogeneous round or oval lesion, occasionally multinodular or with polycyclic lobulated margins (unfrequently inhomogeneous)	Hypervascularized, with arborization and branches	Homogeneously hyperenhanced (rapid wash-in and wash-out)	Hard lesions with low/absent elastic strain
Embryonal cell carcinoma	Testicular swelling, pain, lumbar pain; palpable firm testicular mass; possible gynecomastia	Can be positive α-FP, β-hCG, LDH(not always)	Hypoechoic heterogeneous lesions with irregular polylobate margins;can present internal cystic areas or calcific margins.	Hypervascularized/avascular	Enhanced/unenhanced ORperilesional rim enhancement	Hard lesions with low/absent elastic strain
Teratoma	Testicular swelling, pain, lumbar pain; palpable firm testicular mass; possible gynecomastia	Can be positive α-FP, β-hCG, LDH(not always)	Heterogeneous lesions, well-circumscribed with cystic areas and internal septa	Hypervascularized in the solid part	Inhomogeneously hyperenhanced	Hard lesions with low/absent elastic strain (depending on liquid amount)
Choriocarcinoma	Testicular swelling, pain, lumbar pain; palpable firm testicular mass; possible gynecomastia	Can be positive β-hCG, (not always)	Heterogeneous lesions with hypo-anechoic areas (hemorrhage, necrosis) and calcifications	Hypervascularized	Hyperenhanced	Hard lesions with low/absent elastic strain
Yolk sac tumors	Testicular swelling, pain, lumbar pain; palpable firm testicular mass	Can be positive α-FP (not always)	Heterogeneous lesions with anechoic areas	Hypervascularized	Hyperenhanced	Hard lesions with low/absent elastic strain
Mixed	Testicular swelling, pain, lumbar pain; palpable firm testicular mass; possible gynecomastia	Can be positive α-FP, β-hCG, LDH(not always)	Different aspect regarding main histological component	Hypervascularized	Homogeneously/ inhomogeneously hyperenhanced	Hard lesions with low/absent elastic strain
Burned-out tumor	Lumbar pain, vomit; possible gynecomastia	Can be positive α-FP, β-hCG, LDH(not always)	No testicular nodule; highly echogenic foci or gross calcifications/hypoechoic irregular areas	Hypovascularized	Unenhanced	/
Lymphoma	Testicular swelling, pain, and specific lymphoma symptoms; affects men older than 50 years, palpable firm testicular mass	Negative	Hypoechoic lesions with diffuse infiltration or multifocal hypoechoic lesions of various size	Hypervascularized with linear non-branching pattern	Hyperenhanced	Hard lesions with low/absent elastic strain
Leukemia	More frequent in children and young patients; it can be asymptomatic	Negative	Infiltrating pattern with irregular hypoechoic longitudinal striae/focal pattern with irregular hypoechoic nodules	Hypervascularized	Inhomogeneously hyperenhanced	Hard lesions with low/absent elastic strain

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
