# Peer review of "Multiparametric Ultrasound for Diagnosing Testicular Lesions: Everything You Need to Know in Daily Clinical Practice"

_cancers, 2023, doi:10.3390/cancers15225332_

Round 1

Reviewer 1 Report

Comments and Suggestions for Authors

The article provides a careful summary of methods for detecting testicular lesions. LMP-US can provide a more detailed characterization of testicular lesions than single ultrasound techniques. I believe that the work can be useful to andrologists who visit them daily and therefore I express a positive opinion on the publication of the work

Author Response

We sincerely thank the Reviewer for his/her appreciation. We agree that LMP-US can provide a more detailed characterization of testicular lesions and should be performed more extensively.

Reviewer 2 Report

Comments and Suggestions for Authors

General comment

The manuscript entitled “Multiparametric ultrasound for diagnosing testicular lesions: everything you need to know in daily clinical practice” by Pozza et al., aims to summarize in a narrative and pictorial review, iconographic characteristics of several benign and malignant testicular lesions in order to provide a helpful guide in the common clinical practice. Overall, the manuscript is well written and very comprehensive of several testicular lesions, posing particular attention to the ultrasound characteristics of different entities. Nevertheless, the manuscript lacks a proper discussion in addition to a few data regarding epidemiology and risk factors of testicular lesions. Considering the title, i.e. “Everything you need to know in daily clinical practice”, a summary of current studies as well as real-life data could be a nice addition. Few other minor corrections are reported followingly.

INTRODUCTION

71-79: redundant, rewrite

85-89: In this scenario, the role of the US is crucial both in diagnosis and follow-up. Especially considering the risk factors associated with testicular cancer, the possibility of improving the diagnostic capability of the US could represent one of the main points of the study. Regarding the risk factors associated with testicular cancer also see: DOI: 10.3390/medicina59040724

METHODS

99: Considering it is a narrative review, this section should be revised accordingly (adding further details on data retrieved) – or deleted.

DIAGNOSTICS STEPS TO PERFORM BEFORE RUNNING mp-US

113: This paragraph could be enriched with epidemiological data about testicular cancer. The previously suggested reference could also be used.

TABLES

Table 1: Check the style of the table in order to provide a more elegant visualization of the data.

Table 2: As before

NEOPLASTIC TESTICULAR LESIONS

763: Summarize, too long compared to other sections.

CONCLUSIONS

924: Before proceeding to the conclusions, I think a brief discussion about the advantages, limitations and future perspective related to the role of multiparametric US should be written. In particular, considering that you reported in the methods a thorough search in the literature, it should be a nice addition to reporting real-life data retrieved from clinical studies. Lastly, I would add the ongoing studies, if available, regarding the topic.

Comments on the Quality of English Language

Minor typos. Overall, few english grammar errors revisable during editing.

Author Response

The manuscript entitled “Multiparametric ultrasound for diagnosing testicular lesions: everything you need to know in daily clinical practice” by Pozza et al., aims to summarize in a narrative and pictorial review, iconographic characteristics of several benign and malignant testicular lesions in order to provide a helpful guide in the common clinical practice. Overall, the manuscript is well written and very comprehensive of several testicular lesions, posing particular attention to the ultrasound characteristics of different entities. Nevertheless, the manuscript lacks a proper discussion in addition to a few data regarding epidemiology and risk factors of testicular lesions. Considering the title, i.e. “Everything you need to know in daily clinical practice”, a summary of current studies as well as real-life data could be a nice addition. Few other minor corrections are reported followingly.

We sincerely thank the Reviewer for his/her appreciation and insightful criticisms on our work.

Since the manuscript is focused on “multiparametric ultrasound for diagnosing testicular lesions”, and “everything you need to know in daily clinical practice” is referred to the ultrasound characteristics of testicular lesions, a detailed digression regarding epidemiology and risk factors of testicular lesions would have been unfocused. However, the main risk factors related to testicular cancer are reported in the new paragraph 2 of the corrected manuscript (lines 127-130) as well as the new reference suggested below DOI: 10.3390/medicina59040724. In addition, a brief discussion about epidemiology of testicular lesions is reported point by point for each lesion and, regarding testicular cancers, in the new paragraph 5 (“Neoplastic testicular lesions”) of the corrected manuscript.

Of note, we have approached all the issues raised by Reviewer 2. We further address each issue in a point-by-point rebuttal below. 

INTRODUCTION

Q1: 71-79: redundant, rewrite

R1. We thank the Reviewer for helping us to clarify this sentence. We deleted unnecessary sentences.

Q2: 85-89: In this scenario, the role of the US is crucial both in diagnosis and follow-up. Especially considering the risk factors associated with testicular cancer, the possibility of improving the diagnostic capability of the US could represent one of the main points of the study. Regarding the risk factors associated with testicular cancer also see: DOI: 10.3390/medicina59040724

R2: We thank the Reviewer for his/her observation. In the new version of the manuscript he main risk factors related to testicular cancer are reported in the new paragraph 2 of the corrected manuscript (lines 127-130) as well as the new reference suggested DOI: 10.3390/medicina59040724,  that is significant for the topic.

METHODS

Q3: 99: Considering it is a narrative review, this section should be revised accordingly (adding further details on data retrieved) – or deleted.

R3: We thank the Reviewer for the comment. Considering that this is a narrative and not a systematic review, according to the Reviewer suggestion, in the corrected version of the manuscript this section has been deleted.

DIAGNOSTICS STEPS TO PERFORM BEFORE RUNNING mp-US

Q4: 113: This paragraph could be enriched with epidemiological data about testicular cancer. The previously suggested reference could also be used. 

R4: We thank the Reviewer for his/her comment. In the new version of the manuscript, a brief discussion about epidemiology of testicular lesions is reported point by point for each lesion and, regarding testicular cancers, in the new paragraph 5 (“Neoplastic testicular lesions”).

The paragraph commented by the Reviewer (new paragraph 2) wants to underlie that clinical history and physical examination are crucial and mandatory after discovering a testicular lesion and before performing mp-US, since they can suggest the diagnostic procedure to undertake. In order to clarify this point, in the new version of the manuscript we modified the title of this paragraph, replacing” Diagnostic steps to perform before running mp-US” with “Brief summary of what to investigate before running mp-US”. In addition, according to the Reviewer suggestion, we added to Paragraph 5 the aforementioned reference DOI: 10.3390/medicina59040724.

TABLES

Q5: Table 1: Check the style of the table in order to provide a more elegant visualization of the data.

       Table 2: As before

R5: We thank the reviewer for the suggestion. A more elegant visualization of the data is now available. The Journal Editorial Office will define the style before eventual publication.

NEOPLASTIC TESTICULAR LESIONS

Q6: 763: Summarize, too long compared to other sections.

R6: We thank the Reviewer for the observation. We agree that this paragraph is longer than  other sections, however the topic should be carefully introduced. According to the Reviewer request, in order to shorten the paragraph, in the new version of the manuscript we deleted some redundant sentences.

CONCLUSIONS

Q7: 924: Before proceeding to the conclusions, I think a brief discussion about the advantages, limitations and future perspective related to the role of multiparametric US should be written. In particular, considering that you reported in the methods a thorough search in the literature, it should be a nice addition to reporting real-life data retrieved from clinical studies. Lastly, I would add the ongoing studies, if available, regarding the topic.

R7: We thank the Reviewer for the comment. As requested, a brief discussion about the advantages, limitations and future perspectives related to the role of multiparametric US has been added to the new version of the manuscript.

Regarding real-life data retrieved from clinical studies, in our opinion the present manuscript represents already a practical overview including also real-life data.

Regarding ongoing studies on the topic, it is possible to find in internet websites related to genetic diagnostics- or to medications-related clinical trials (see, for example, https://www.centerwatch.com/clinical-trials/listings/condition/38/testicular-cancer/), but there is not an available webpage reporting ongoing studies on testicular ultrasound.

Comments on the Quality of English Language

Q8: Minor typos. Overall, few english grammar errors revisable during editing.

R8: We thank the Reviewer for the observation. The grammar errors have been identified and corrected.

Reviewer 3 Report

Comments and Suggestions for Authors

Review Multiparametric ultrasound for diagnosing testicular lesions: 2 everything you need to know in daily clinical practice.

Thank you for the opportunity to review this interesting and important paper

Overall, a very useful review.

Introduction

This in incorrect. Please rephrase this sentence, page 2, line 86 “…US-related risk factors for malignancy ( E.g., cryptorchidism and microlithiasis). Perhaps you mean microlithiasis? Testicular microlithiasis is a suggested association, but no cause-and-effect studies exist. Many new studies – published in the last decade – also acknowledge this. Perhaps include some of the newest references?

Methods

The identification of relevant studies in the English language was performed by the authors – Was it performed by all or just some of the authors?

Who decided to include or exclude papers? What happens if case of different opinion about inclusion?

Can you provide a table for an overview?

During which period was the screening of the paper performed?

This section is very limited.

Results

Table 1 +2 provides a fulfilling overview.

Page 4, line 148. The volume of testes – do you recommend a specific volume formula?

The sections on the specific Ultrasound techniques are very long. Keep it simple. Less is more…

The figures. Please include more patient information. E.g. Patient age, symptoms, and year of investigation.

For a better overview, the images could be placed directly after the relevant description of the condition.

Author Response

Thank you for the opportunity to review this interesting and important paper.

Overall, a very useful review.

We sincerely thank the Reviewer for his/her appreciation of our work. 

Introduction

Q1: This in incorrect. Please rephrase this sentence, page 2, line 86 “…US-related risk factors for malignancy ( E.g., cryptorchidism and microlithiasis). Perhaps you mean microlithiasis? Testicular microlithiasis is a suggested association, but no cause-and-effect studies exist. Many new studies – published in the last decade – also acknowledge this. Perhaps include some of the newest references?

R1: We thank Reviewer #3 for his valuable comment and suggestion. We absolutely agree with him/her.

Recent meta-analyses support a significant association between TML and testicular cancer in the general male population (1) and children with contributing factors for primary testicular tumor (2). However, recent reviews (3,4), reported that TML is not an independent risk factor for testicular cancer but is associated with malignancy depending on the co-occurrence of specific risky conditions. This is why we wrote in the text “cryptorchidism AND microlithiasis”. Hence, we agree with the Reviewer that the topic “microlithiasis and testicular malignancy” is still debated.

Anyway, since Reviewer #2 asked to rephrase that part of the introduction as it was redundant, and in order to avoid an incorrect information, in the new version of the manuscript we deleted that sentence.

  1. Wang T,Liu LLuo JLiu TWei AA meta-analysis of the relationship between testicular microlithiasis and incidence of testicular cancerUrol J. 2015; 12(2): 2057-2064.
  2. Yu CJ,Lu JDZhao J, et al. Incidence characteristics of testicular microlithiasis and its association with risk of primary testicular tumors in children: a systematic review and meta-analysisWorld J Pediatr. 2020; 16(6): 585-597.
  3. Pedersen MR,Rafaelsen SRMøller HVedsted POsther PJTesticular microlithiasis and testicular cancer: review of the literatureInt Urol Nephrol. 2016; 48(7): 1079-1086.
  4. Balawender K,Orkisz SWisz PTesticular microlithiasis: what urologists should know. A review of the current literatureCent European J Urol. 2018; 71(3): 310-314.

Methods

Q2: The identification of relevant studies in the English language was performed by the authors – Was it performed by all or just some of the authors?

Who decided to include or exclude papers? What happens if case of different opinion about inclusion?

Can you provide a table for an overview?

During which period was the screening of the paper performed?

This section is very limited.

R2: We thank the Reviewer #3 for the comment. According to the Reviewer #2 request, in the corrected version of the manuscript the Methods section has been deleted, being this a narrative and not a systematic review.

Results

Q3: Table 1 +2 provides a fulfilling overview.

R3: We thank the Reviewer #3 for his comment.

Q4: Page 4, line 148. The volume of testes – do you recommend a specific volume formula?

R4: We thank the Reviewer for his/her question. Various formulas have been used in the US assessment of testicular volume: the ellipsoid (L × W × H × 0.52), the prolate spheroid (L × W × W x 0.52) and the empirical Lambert’s (L × W × H × 0.71) formulas. We suggest using the ellipsoid formula for adult testes, according to the European Academy of Andrology (EAA) suggestion (1) while the Lambert’s formula could be preferred in pre-pubertal testes (2). In the new version of the manuscript we clarified this point in the text (lines 156-158).

        1.Lotti, F.; Frizza, F.; Balercia, G.; Barbonetti, A.; Behre, H.M.; Calogero, A.E.; Cremers, J.F.; Francavilla, F.; Isidori, A.M.; Kliesch, S.; et al. The European Academy of Andrology (EAA) Ultrasound Study on Healthy, Fertile Men: Scrotal Ultrasound Reference Ranges and Associations with Clinical, Seminal, and Biochemical Characteristics. Andrology 2021, 9, 559–576, doi:10.1111/andr.12951.

        2.Isidori, A.M.; Lenzi, A. Ultrasound of the Testis for the Andrologist: Morphological and Functional Atlas; Springer, 2018; ISBN 9783319518268.

Q5: The sections on the specific Ultrasound techniques are very long. Keep it simple. Less is more…

R5: According to the Reviewer #3 request the sections on the specific ultrasound techniques have been shortened.

Q6: The figures. Please include more patient information. E.g. Patient age, symptoms, and year of investigation.

R6: We thank Reviewer #3 for his valuable suggestion. We improved the figure’s captions adding patient’s information in the figure legends.

For a better overview, the images could be placed directly after the relevant description of the condition.

We thank the Referee for the comment. We will ask to the Journal Editorial Team to follow this suggestion.

Round 2

Reviewer 2 Report

Comments and Suggestions for Authors

The authors improved the manuscript accordingly to previous suggestions. No further corrections required.

Comments on the Quality of English Language

None